# ON THE CONVERGENCE OF ADAGRAD(NORM) ON $\mathbb{R}^d$: BEYOND CONVEXITY, NON-ASYMPTOTIC RATE AND ACCELERATION

**Zijian Liu**[*]
New York University
zl3067@nyu.edu

**Ta Duy Nguyen**[*] **& Alina Ene**
Boston University
{taduy,aene}@bu.edu

**Huy L. Nguyen**
Northeastern University
hu.nguyen@northeastern.edu

## ABSTRACT

Existing analysis of AdaGrad and other adaptive methods for smooth convex optimization is typically for functions with bounded domain diameter. In unconstrained problems, previous works guarantee an asymptotic convergence rate without an explicit constant factor that holds true for the entire function class. Furthermore, in the stochastic setting, only a modified version of AdaGrad, different from the one commonly used in practice, in which the latest gradient is not used to update the stepsize, has been analyzed. Our paper aims at bridging these gaps and developing a deeper understanding of AdaGrad and its variants in the standard setting of smooth convex functions as well as the more general setting of quasar convex functions. First, we demonstrate new techniques to explicitly bound the convergence rate of the vanilla AdaGrad for unconstrained problems in both deterministic and stochastic settings. Second, we propose a variant of AdaGrad for which we can show the convergence of the last iterate, instead of the average iterate. Finally, we give new accelerated adaptive algorithms and their convergence guarantee in the deterministic setting with explicit dependency on the problem parameters, improving upon the asymptotic rate shown in previous works.

## 1 INTRODUCTION

In recent years, the prevalence of machine learning models has motivated the development of new optimization tools, among which adaptive methods such as Adam (Kingma & Ba, 2014), AmsGrad (Reddi et al., 2018), AdaGrad (Duchi et al., 2011) emerge as the most important class of algorithms. These methods do not require the knowledge of the problem parameters when setting the stepsize as traditional methods like SGD, while still showing robust performances in many ML tasks.

However, it remains a challenge to analyze and understand the properties of these methods. Take AdaGrad and its variants for example. In its vanilla scalar form, also known as AdaGradNorm, the step size is set using the cumulative sum of the gradient norm of all iterates so far. The work of Ward et al. (2020) has shown the convergence of this algorithm for non-convex funtions by bounding the decay of the gradient norms. However, in convex optimization, usually we require a stronger convergence criterion—bounding the function value gap. This is where we lack theoretical understanding. Even in the deterministic setting, most existing works (Levy, 2017; Levy et al., 2018; Ene et al., 2021) rely on the assumption that the domain of the function is bounded. The dependence on the domain diameter can become an issue if it is unknown or cannot be readily estimated. Other works for unconstrained problems (Antonakopoulos et al., 2020; 2022) offer a convergence rate that depends on the limit of the step size sequence. This limit is shown to exist for each function, but without an explicit value, and more importantly, it is not shown to be a constant for the entire function class. This means that these methods essentially do not tell us how fast the algorithm converges in the worst case. Another work by Ene & Nguyen (2022) gives an explicit rate of convergence for the entire class but requires the strong assumption that the gradients are bounded even in the smooth setting and the convergence guarantee has additional error terms depending on this bound.

---

[*]Equal contribution, corresponding authors.

In the stochastic setting, one common approach is to analyze a modified version of AdaGrad with off-by-one step size, i.e. the gradient at the current time step is not taken into account when setting the new step size. This is where the gap between theory and practice exists.

## 1.1 OUR CONTRIBUTION

In this paper, we make the following contributions. First, we demonstrate a method to show an explicit non-asymptotic convergence rate of AdaGradNorm and AdaGrad on $\mathbb{R}^d$ in the deterministic setting. Our method extends to a more general function class known as $\gamma$-quasar convex functions with a weaker condition for smoothness. To the best of our knowledge, we are the first to prove this result. Second, we present new techniques to analyze stochastic AdaGradNorm and offer an explicit convergence guarantee for $\gamma$-quasar convex optimization on $\mathbb{R}^d$ with a mild assumption on the noise of the gradient estimates. We propose two new variants of AdaGradNorm which demonstrate the convergence of the last iterate instead of the average iterate as shown in AdaGradNorm. Finally, we propose a new accelerated algorithm with two variants and show their non-asymptotic convergence rate in the deterministic setting.

## 1.2 RELATED WORK

**Adaptive methods** There has been a long line of works on adaptive methods, including AdaGrad (Duchi et al., 2011), RMSProp (Tieleman et al., 2012) and Adam (Kingma & Ba, 2014). AdaGrad was first designed for stochastic online optimization; subsequent works (Levy, 2017; Kavis et al., 2019; Bach & Levy, 2019; Antonakopoulos et al., 2020; Ene et al., 2021) analyzed AdaGrad and various adaptive algorithms for convex optimization and generalized them for variational inequality problems. These works commonly assume that the optimization problem is contrained in a set with bounded diameter. Li & Orabona (2019) are the first to analyze a variant of AdaGrad for unbounded domains where the latest gradient is not used to construct the step size, which differs from the standard version of AdaGrad commonly used in practice. However, the algorithm and analysis of Li & Orabona (2019) set the initial step size based on the smoothness parameter and thus they do not adapt to it. Other works provide convergence guarantees for adaptive methods for unbounded domains, yet without explicit dependency on the problem parameters (Antonakopoulos et al., 2020; 2022), or for a class of strongly convex functions (Xie et al., 2020). Another work by Ene & Nguyen (2022) requires the strong assumption that the gradients are bounded even for smooth functions and the convergence guarantee has additional error terms depending on the gradient upperbound. Our work analyzes the standard version of AdaGrad for unconstrained and general convex problems and shows explicit convergence rate in both the deterministic and stochastic setting.

Accelerated adaptive methods have been designed to achieve $O(1/T^2)$ and $O(1/\sqrt{T})$ respectively in the deterministic and stochastic setting in the works of Levy et al. (2018); Ene & Nguyen (2022); Antonakopoulos et al. (2022). We show different variants and demonstrate the same but explicit accelerated convergence rate in the deterministic setting for unconstrained problems.

**Analysis beyond convexity** The convergence of some variants of AdaGrad has been established for nonconvex functions in the work of Li & Orabona (2019); Ward et al. (2020); Faw et al. (2022) under various assumptions. Other works (Li & Orabona, 2020; Kavis et al., 2022) demonstrate the convergence with high probability. We refer the reader to Faw et al. (2022) for a more detailed survey on AdaGrad-style methods for nonconvex optimization. In general, the criterion used to study these convergence rates is the gradient norm of the function, which is weaker than the function value gap normally used in the study of convex functions. In comparison, we study the convergence of AdaGrad via the function value gap for a broader notion of convexity, known as quasar-convexity, as well as a more generalized definition of smoothness.

## 2 PRELIMINARIES

We consider the following optimization problem: $\text{minimize}_{x \in \mathbb{R}^d} F(x)$, where $F$ is differentiable satisfying $F^* = \inf_{x \in \mathbb{R}^d} F(x) > -\infty$ and $x^* \in \arg\min_{x \in \mathbb{R}^d} F(x) \neq \emptyset$. We will use the following notations throughout the paper: $a^+ = \max\{a, 0\}$, $a \vee b = \max\{a, b\}$, $[n] = \{1, 2, \cdots, n\}$, and $\|\cdot\|$ denotes the $\ell_2$-norm $\|\cdot\|_2$ for simplicity.

| **Algorithm 1** AdaGradNorm | **Algorithm 2** Stochastic AdaGradNorm |
|---|---|
| Initialize: $x_1, \eta > 0$ | Initialize: $x_1, \eta > 0$ |
| for $t = 1$ to $T$ | for $t = 1$ to $T$ |
| $\quad b_t = \sqrt{b_0^2 + \sum_{i=1}^{t} \|\nabla F(x_i)\|^2}$ | $\quad b_t = \sqrt{b_0^2 + \sum_{i=1}^{t} \|\widehat{\nabla} F(x_i)\|^2}$ |
| $\quad x_{t+1} = x_t - \frac{\eta}{b_t} \nabla F(x_t)$ | $\quad x_{t+1} = x_t - \frac{\eta}{b_t} \|\widehat{\nabla} F(x_t)\|^2$ |

Additionally, we list below the assumptions that will be used in the paper.

**1. $\gamma$-quasar convexity**: There exists $\gamma \in (0, 1]$ such that $F^* \geq F(x) + \frac{1}{\gamma} \langle \nabla F(x), x^* - x \rangle, \forall x \in \mathbb{R}^d$ where $x^* \in \arg\min_{x \in \mathbb{R}^d} F(x)$. When $\gamma = 1$, $F$ is also known as star-convex.

**1'. Convexity**: $F$ is convex. This stronger assumption implies that Assumption 1 holds with $\gamma = 1$.

**2. Weak $L$-smoothness**: $\exists L > 0$ such that $F(x) - F^* \geq \|\nabla F(x)\|^2/2L, \forall x \in \mathbb{R}^d$.

**2'. $L$-smoothness**: $\exists L > 0$ such that $F(x) \leq F(y) + \langle \nabla F(y), x - y \rangle + \frac{L}{2}\|x - y\|^2, \forall x, y \in \mathbb{R}^d$.

**2''. $L$-smoothness**: $\exists \mathbf{L} = \text{diag}\left(L_{i \in [d]}\right)$ with $L_i > 0$ such that $F(x) \leq F(y) + \langle \nabla F(y), x - y \rangle + \frac{1}{2}\|x - y\|_{\mathbf{L}}^2, \forall x, y \in \mathbb{R}^d$ where $\|a\|_{\mathbf{L}} = \sqrt{\langle a, \mathbf{L}a \rangle}$.

In the stochastic setting, we assume that we have access to a stochastic gradient oracle $\widehat{\nabla} F(x)$ that is independent of the history of the randomness and it satisfies the following assumptions:

**3. Unbiased gradient estimate**: $\mathbb{E}[\widehat{\nabla} F(x)] = \nabla F(x)$.

**4. Sub-Weibull noise**: $\mathbb{E}\left[\exp\left((\|\widehat{\nabla} F(x) - \nabla F(x)\|/\sigma)^{1/\theta}\right)\right] \leq \exp(1)$ for some $\theta > 0$.

Here, we give a brief discussion of our assumptions. Assumption 1 is introduced by Hinder et al. (2020) and it is strictly weaker than Assumption 1'. Assumption 2 is a relaxation of Assumption 2', the latter is the standard definition of smoothness used in many existing works (see Guille-Escuret et al. (2021) for a detailed comparison between different smoothness conditions). Assumption 2'' is used to analyze the AdaGrad algorithm which uses per-coordinate step sizes. Assumption 3 is a standard assumption in stochastic optimization problems. Assumption 4 is more general and encapsulates sub-Gaussian ($\theta = 1/2$, used in Li & Orabona (2019)) and sub-exponential noise ($\theta = 1$). We refer the reader to Vladimirova et al. (2020) for more discussion on sub-Weibull noise.

## 3 CONVERGENCE OF ADAGRADNORM ON $\mathbb{R}^d$ UNDER $\gamma$-QUASAR CONVEXITY

We first turn our attention to AdaGradNorm (Algorithm 1) in the deterministic setting, which will serve as the basis for the understanding of Stochastic AdaGradNorm (Algorithm 2) and deterministic AdaGrad (Algorithm 7). To the best of our knowledge, we are the first to present the explicit convergence rate of these three algorithms on $\mathbb{R}^d$. Due to the space limit, we defer the theorem of the convergence guarantee of AdaGrad and its proof to Section A.3 in the appendix.

### 3.1 ADAGRADNORM

Previous analysis of AdaGradNorm often aims at bounding the gradient norm of smooth nonconvex functions, or is conducted for smooth convex functions in constrained problems with a bounded domain. Bounding the gradient norm is strictly weaker than bounding the function value gap due to the fact that $\|\nabla F(x)\|^2 \leq 2L(F(x) - F^*)$, where $L$ is the smoothness parameter. For convex functions, the common analysis will always meet the following intermediate step

$$F(x_t) - F^* \leq \frac{b_t}{2\eta}\left[\|x_t - x^*\|^2 - \|x_{t+1} - x^*\|^2\right] + \text{Other terms}.$$

Assuming a bounded domain is a way to making the terms $\frac{b_t}{2\eta}\left[\|x_t - x^*\|^2 - \|x_{t+1} - x^*\|^2\right]$ telescope after taking the sum over all iterations $t$. This is critical in the analysis, but at the same

time leads to the dependence on the domain diameter, which can be hard to estimate. For unconstrained problems, a natural approach is to divide the terms by $b_t$, so that the remaining terms $\frac{1}{2\eta}\left[\|x_t - x^*\|^2 - \|x_{t+1} - x^*\|^2\right]$ can telescope. Our key insight is that we can bound the function value gap via the step size $b_t$, which in turn can be bounded via the function value gap. This self-bounding argument allows us to finally prove the convergence rate. This result holds under more general conditions than convexity and smoothness (Assumptions 1 and 2).

**Theorem 3.1.** *With Assumptions 1 and 2, AdaGradNorm admits*

$$\frac{\sum_{t=1}^{T} F(x_t) - F^*}{T} \leq \frac{\left(\frac{2L\|x_1 - x^*\|^2}{\gamma\eta} + \frac{4\eta L}{\gamma}\log^+ \frac{2\eta L}{\gamma b_0} + b_0\right)\left(\frac{\|x_1 - x^*\|^2}{\gamma\eta} + \frac{2\eta}{\gamma}\log^+ \frac{2\eta L}{\gamma b_0}\right)}{T}$$

*Proof.* Starting from the $\gamma$-quasar convexity of $F$, we have

$$F(x_t) - F^* \leq \frac{\langle \nabla F(x_t), x_t - x^* \rangle}{\gamma} = \frac{b_t}{\gamma\eta}\langle x_t - x_{t+1}, x_t - x^* \rangle$$

$$= \frac{b_t}{2\gamma\eta}\left[\|x_t - x^*\|^2 - \|x_{t+1} - x^*\|^2 + \|x_{t+1} - x_t\|^2\right]$$

Notice that $x_{t+1} - x_t = -\eta b_t^{-1}\nabla F(x_t)$. Dividing both sides by $b_t$ and taking the sum over $t$, we obtain

$$\sum_{t=1}^{T} \frac{F(x_t) - F^*}{b_t} \leq \frac{\|x_1 - x^*\|^2}{2\gamma\eta} + \sum_{t=1}^{T} \frac{\eta}{2\gamma b_t^2}\|\nabla F(x_t)\|^2.$$

Note that $F$ also satisfies Assumption 2, i.e., $F(x_t) - F^* \geq \frac{\|\nabla F(x_t)\|^2}{2L}$. Therefore

$$\sum_{t=1}^{T} \frac{F(x_t) - F^*}{2b_t} + \frac{\|\nabla F(x_t)\|^2}{4Lb_t} \leq \sum_{t=1}^{T} \frac{F(x_t) - F^*}{b_t} \leq \frac{\|x_1 - x^*\|^2}{2\gamma\eta} + \sum_{t=1}^{T} \frac{\eta}{2\gamma b_t^2}\|\nabla F(x_t)\|^2$$

$$\Rightarrow \sum_{t=1}^{T} \frac{F(x_t) - F^*}{b_t} \leq \frac{\|x_1 - x^*\|^2}{\gamma\eta} + \underbrace{\sum_{t=1}^{T}\left(\frac{\eta}{\gamma b_t^2} - \frac{1}{2Lb_t}\right)\|\nabla F(x_t)\|^2}_{A}.$$

We can bound the term $A$ by the technique commonly used in the analysis of adaptive methods. Let $\tau$ be the last $t$ such that $b_t \leq \frac{2\eta L}{\gamma}$. If $b_1 > \frac{2\eta L}{\gamma}$, we have $A < 0 \leq \frac{2\eta}{\gamma}\log^+ \frac{2\eta L}{\gamma b_0}$. Otherwise

$$A \leq \sum_{t=1}^{\tau}\left(\frac{\eta}{\gamma b_t^2} - \frac{1}{2Lb_t}\right)\|\nabla F(x_t)\|^2 \leq \sum_{t=1}^{\tau} \frac{\eta}{\gamma}\frac{b_t^2 - b_{t-1}^2}{b_t^2} \leq \frac{\eta}{\gamma}\sum_{t=1}^{\tau}\log \frac{b_t^2}{b_{t-1}^2} \leq \frac{2\eta}{\gamma}\log^+ \frac{2\eta L}{\gamma b_0}.$$

Thus we always have $A \leq \frac{2\eta}{\gamma}\log^+ \frac{2\eta L}{\gamma b_0}$, and obtain

$$\sum_{t=1}^{T} \frac{F(x_t) - F^*}{b_t} \leq \frac{\|x_1 - x^*\|^2}{\gamma\eta} + \frac{2\eta}{\gamma}\log^+ \frac{2\eta L}{\gamma b_0},$$

which gives

$$\sum_{t=1}^{T} F(x_t) - F^* \leq b_T\left(\frac{\|x_1 - x^*\|^2}{\gamma\eta} + \frac{2\eta}{\gamma}\log^+ \frac{2\eta L}{\gamma b_0}\right).$$

Note that by Assumption 2 again, we have

$$b_T = \sqrt{b_0^2 + \sum_{t=1}^{T}\|\nabla F(x_t)\|^2} \leq \sqrt{b_0^2 + \sum_{t=1}^{T} 2L\left(F(x_t) - F^*\right)}.$$

Let $\Delta_T = \sum_{t=1}^{T} F(x_t) - F^*$, then

$$\Delta_T \leq \sqrt{b_0^2 + 2L\Delta_T}\left(\frac{\|x_1 - x^*\|^2}{\gamma\eta} + \frac{2\eta}{\gamma}\log^+ \frac{2\eta L}{\gamma b_0}\right)$$

$$\Rightarrow \Delta_T \leq \left(\frac{2L\|x_1 - x^*\|^2}{\gamma\eta} + \frac{4\eta L}{\gamma}\log^+ \frac{2\eta L}{\gamma b_0} + b_0\right)\left(\frac{\|x_1 - x^*\|^2}{\gamma\eta} + \frac{2\eta}{\gamma}\log^+ \frac{2\eta L}{\gamma b_0}\right).$$

Dividing both sides by $T$, we get the desired result. $\square$

When $F$ is convex (which implies $\gamma = 1$), using the above theorem and convexity, we obtain the following convergence rate for the average iterate:

**Corollary 3.2.** *With Assumptions 1' and 2, for $\bar{x}_T = \frac{\sum_{t=1}^{T} x_t}{T}$, AdaGradNorm admits*

$$F(\bar{x}_T) - F^* \leq \frac{\left( \frac{2L\|x_1 - x^*\|^2}{\eta} + 4\eta L \log^+ \frac{2\eta L}{b_0} + b_0 \right) \left( \frac{\|x_1 - x^*\|^2}{\eta} + 2\eta \log^+ \frac{2\eta L}{b_0} \right)}{T}.$$

The rate in Theorem 3.1 can be improved by a factor $1/\gamma$ by replacing Assumption 2 by 2'. The details and the proof are deferred into Section A.1 in the appendix.

## 3.2 Stochastic AdaGradNorm

In this section, we consider the stochastic setting where we only have access to an unbiased gradient estimate $\widehat{\nabla} F(x_t)$ of $\nabla F(x_t)$ (Assumption 3). As expected for a stochastic method, the accumulation of noise is the reason that we can only expect an $O(1/\sqrt{T})$ convergence rate, instead of $O(1/T)$. This convergence rate is already shown by prior works (Levy et al., 2018) under the bounded domain assumption. However, in an unbounded domain, when extending our previous analysis to the stochastic setting, that is, dividing both sides by $b_t$, we will face several challenges. One of such is the term $b_t^{-1} \langle \nabla F(x_t) - \widehat{\nabla} F(x_t), x_t - x^* \rangle$. To handle this term, often we see that existing works, such as Li & Orabona (2019), analyze a modified version of Stochastic AdaGradNorm with off-by-one stepsize, i.e., $b_t = \sqrt{b_0^2 + \sum_{i=1}^{t-1} \|\widehat{\nabla} F(x_i)\|^2}$ in which the latest gradient $\widehat{\nabla} F(x_t)$ is not used to calculate $b_t$. This allows to decouple the dependency of $b_t$ on the randomness at time $t$, thus in expectation $\mathbb{E}[b_t^{-1} \langle \nabla F(x_t) - \widehat{\nabla} F(x_t), x_t - x^* \rangle] = 0$. Yet, this analysis does not apply to the standard algorithm which is more commonly used in practice.

To the best of our knowledge, we are the first to propose a new technique that can show the convergence of Algorithm 2 on $\mathbb{R}^d$ without going through the off-by-one stepsize. Here, we briefly compared the assumptions in our analysis with the assumptions in Li & Orabona (2019). Assumptions 2' and 3 used in both works are standard. Meanwhile, Assumptions 1 ($\gamma$-quasar convexity) and 4 (sub-Weibull noise) in our analysis are much weaker than the convexity and sub-Gaussian noise assumptions in Li & Orabona (2019). Besides, we note that, while the guarantee in Li & Orabona (2019) is a bound on $\mathbb{E}\left[ \sqrt{(\sum_{t=1}^{T} F(x_t) - F(x^*))/T} \right]$, we will present a bound for $\mathbb{E}\left[ (\sum_{t=1}^{T} F(x_t) - F(x^*))/T \right]$, which is a stronger criterion that is often used in convex analysis. We also remark that the algorithm and analysis of Li & Orabona (2019) still require the smoothness parameter to set the initial stepsize, thus their method is not fully adaptive.

The first observation is that, if we let $\xi_t := \widehat{\nabla} F(x_t) - \nabla F(x_t)$ be the stochastic error and $M_T := \max_{t \in [T]} \|\xi_t\|^2$, $M_T$ is bounded by $\sigma^2 \log^{2\theta} \frac{eT}{\delta}$ with probability at least $1 - \delta$ (c.f. Lemma A.4 in Appendix A), which can give a high probability bound on $b_T$.

**Lemma 3.3.** *Suppose $F$ satisfies Assumptions 2' and 4, if $M_T \leq \sigma^2 \log^{2\theta} \frac{eT}{\delta}$, then*

$$b_T \leq 2b_0 + \frac{4(F(x_1) - F^*)}{\eta} + 4\eta L \log^+ \frac{\eta L}{b_0} + 4\sigma \sqrt{T \log^{2\theta} \frac{eT}{\delta} \log \left( 1 + \frac{16\sigma^2 T \log^{2\theta} \frac{eT}{\delta}}{b_0^2} \right)}.$$

Lemma 3.3 gives us an insight: $b_t = \widetilde{O}(1 + \sigma \sqrt{t \log^{2\theta} t})$. Note that this can be expected since we know the classic choice of the step size for SGD is of the order of $O(1 + \sigma \sqrt{t})$. Hence, if we are willing to accept extra $\log$ terms in the convergence guarantee, the appearance of $\log b_t$ is accommodatable. Next we will introduce our novel technique, which, to the best of our knowledge, is the first method that allows us to analyze the standard Stochastic AdaGradNorm on $\mathbb{R}^d$.

**Lemma 3.4.** *Suppose $F$ satisfies Assumptions 1 and 3 then*

$$\mathbb{E}\left[ \frac{\sum_{t=1}^{T} F(x_t) - F(x^*)}{b_T} \right] \leq \frac{\|x_1 - x^*\|^2}{\gamma \eta} + \frac{2\eta}{\gamma} \mathbb{E}\left[ \frac{M_T}{b_0^2} + \log \frac{b_T}{b_0} \right]. \tag{1}$$

*Proof sketch.* Starting from the $\gamma$-quasar convexity, with simple transformations, we obtain

$$F(x_t) - F^* \leq \frac{\langle -\xi_t, x_t - x^* \rangle}{\gamma} + \frac{b_t}{2\gamma\eta} \left( \|x_t - x^*\|^2 - \|x_{t+1} - x^*\|^2 + \|x_{t+1} - x_t\|^2 \right).$$

Here we introduce our novel technique: instead of dividing by $b_t$, we divide both sides by $2b_t - b_0$. This divisor causes a slight non-uniformity between the coefficients of the distance terms $\|x_t - x^*\|^2$ making the sum of them not telescoping. However, this is exactly what we want to handle the difficult term $\frac{\langle -\xi_t, x_t - x^* \rangle}{\gamma(2b_t - b_0)}$ which does not disappear after taking the expectation.

$$\mathbb{E}\left[ \frac{F(x_t) - F^*}{2b_t - b_0} \right] \leq \mathbb{E}\left[ \frac{\langle -\xi_t, x_t - x^* \rangle}{\gamma(2b_t - b_0)} + \frac{b_t}{2b_t - b_0} \times \frac{\|x_t - x^*\|^2 - \|x_{t+1} - x^*\|^2 + \|x_{t+1} - x_t\|^2}{2\gamma\eta} \right].$$

The key step is to use Cauchy-Schwarz inequality for the term

$$|\langle -\xi_t, x_t - x^* \rangle| \leq \frac{\lambda}{2} \|\xi_t\|^2 + \frac{1}{2\lambda} \|x_t - x^*\|^2$$

with the appropriate coefficient $\lambda$ so that the term $\|x_t - x^*\|^2$ can be absorbed to make a telescoping sum $\frac{b_{t-1}\|x_t - x^*\|^2}{2b_{t-1} - b_0} - \frac{b_t\|x_{t+1} - x^*\|^2}{2b_t - b_0}$. The remaining terms are free of $x^*$; hence can be more easily bounded. We can obtain

$$\mathbb{E}\left[ \frac{F(x_t) - F^*}{2b_t - b_0} \right] \leq \mathbb{E}\left[ Z_t \|\xi_t\|^2 + \frac{b_{t-1}\|x_t - x^*\|^2}{2\gamma\eta(2b_{t-1} - b_0)} - \frac{b_t\|x_{t+1} - x^*\|^2}{2\gamma\eta(2b_t - b_0)} + \frac{\eta \left\|\widehat{\nabla} F(x_t)\right\|^2}{2\gamma b_t^2} \right],$$

where $Z_t = \frac{\eta}{\gamma b_0} \left( \frac{1}{2b_{t-1} - b_0} - \frac{1}{2b_t - b_0} \right)$. Now we have a telescoping sum $\frac{b_{t-1}\|x_t - x^*\|^2}{2\gamma\eta(2b_{t-1} - b_0)} - \frac{b_t\|x_{t+1} - x^*\|^2}{2\gamma\eta(2b_t - b_0)}$. Taking the sum over $t$, we have

$$\mathbb{E}\left[ \sum_{t=1}^{T} \frac{F(x_t) - F^*}{2b_t} \right] \leq \mathbb{E}\left[ \sum_{t=1}^{T} \frac{F(x_t) - F^*}{2b_t - b_0} \right]$$

$$\leq \frac{\|x_1 - x^*\|^2}{2\gamma\eta} + \mathbb{E}\left[ \sum_{t=1}^{T} Z_t \|\xi_t\|^2 \right] + \mathbb{E}\left[ \sum_{t=1}^{T} \frac{\eta \left\|\widehat{\nabla} F(x_t)\right\|^2}{2\gamma b_t^2} \right].$$

Proceeding to bound each term, we will obtain Lemma 3.4. $\qquad\square$

We emphasize the following crucial aspect of Lemma 3.4: the inequality gives us a relationship between the function gap and the stepsize $b_T$, which we know how to bound with high probability under Assumptions 2' and 4. On the other hand, this relationship is not ideal due to the fact that on the L.H.S. of (1), we have not obtained a decoupling between the function gap and $b_T$. To this end, we introduce the second novel technique. Let $\Delta_T := \sum_{t=1}^{T} F(x_t) - F(x^*)$, we write

$$\Delta_T = \Delta_T \mathbb{1}_{E(\delta)} + \Delta_T \mathbb{1}_{E^c(\delta)}$$

where we define the event $E(\delta) = \left\{ M_T \leq \sigma^2 \log^{2\theta} \frac{eT}{\delta} \right\}$. For the first term, when $E(\delta)$ happens, we also know from Lemma 3.3 that the stepsize is bounded. Thus we can bound

$$\mathbb{E}\left[ \Delta_T \mathbb{1}_{E(\delta)} \right] = \mathbb{E}\left[ \frac{\sum_{t=1}^{T} F(x_t) - F(x^*)}{b_T} b_T \mathbb{1}_{E(\delta)} \right]$$

which leads us back to Lemma 3.4. We can bound the second term using a tail bound for the event $E^c(\delta)$, knowing from the first observation that $\Pr\left[ E^c(\delta) \right] \leq \delta$. From this insight, and using the self-bounding argument as in the proof of Theorem 3.1, we finally obtain the following result.

**Theorem 3.5.** *Suppose $F$ satisfies Assumptions 1, 2', 3 and 4, Stochastic AdaGradNorm admits*

$$\mathbb{E}\left[ \frac{\sum_{t=1}^{T} F(x_t) - F(x^*)}{T} \right] = O\left( \left( 1 + \text{poly}\left( \sigma^2 \log^{2\theta} T, \log(1 + \sigma^2 T \log^{2\theta} T) \right) \right) \left( \frac{1}{T} + \frac{\sigma \log^\theta T}{\sqrt{T}} \right) \right).$$

---

**Algorithm 3** AdaGradNorm-Last

Initialize: $x_1, \eta > 0, \Delta > 0, p_t > 0$
for $t = 1$ to $T$

$$b_t = \left(b_0^{2+\Delta} + \sum_{i=1}^t \frac{\|\nabla F(x_i)\|^2}{p_i}\right)^{\frac{1}{2+\Delta}}$$

$$x_{t+1} = x_t - \frac{\eta}{b_t}\nabla F(x_t)$$

---

**Algorithm 4** AdaGradNorm-Last

Initialize: $x_1, \eta > 0, \delta \in [2/3, 1), p_t > 0$
for $t = 1$ to $T$

$$b_t = \left(b_0^2 + \sum_{i=1}^t \frac{\|\nabla F(x_i)\|^2}{p_i}\right)^{\frac{1}{2}}$$

$$x_{t+1} = x_t - \frac{\eta}{b_t^\delta b_{t-1}^{1-\delta}}\nabla F(x_t)$$

---

*Remark* 3.6. In the big-$O$ notation, we only show the dependency on $\sigma, T$ and $\theta$ for simplicity. The dependency on the other parameters will be made explicit in the proof of the theorem. By setting $\sigma = 0$, we obtain the standard convergence rate $\mathbb{E}\left[(\sum_{t=1}^T F(x_t) - F(x^*))/T\right] = O(1/T)$ as shown in Section 3.1 for the deterministic setting. This means our analysis adapts to the noise parameter $\sigma$.

Finally, it is worth pointing out that even when we relax Assumption 2' to Assumption 2, we can still provide a convergence guarantee for Stochastic AdaGradNorm. We present the result in Theorem A.9 in the appendix.

## 4 LAST ITERATE CONVERGENCE OF VARIANTS OF ADAGRADNORM FOR $\gamma$-QUASAR CONVEX AND SMOOTH MINIMIZATION ON $\mathbb{R}^d$

In Section 3, under Assumptions 1 and 2, we proved that the average iterate produced by AdaGrad-Norm converges at the $1/T$ rate, i.e., $\left(\sum_{t=1}^T F(x_t) - F^*\right)/T = O(1/T)$. A natural question is whether there exists an adaptive algorithm that can guarantee the convergence of the last iterate. In this section, we give an affirmative answer by presenting two simple variants of AdaGradNorm and show convergence of the last iterate under Assumptions 1 and 2'.

In Algorithm 3, by setting $p_i = i^{-1}$, $\|\nabla F(x_i)\|^2$ has a bigger coefficient than in the standard AdaGradNorm. Should we use the $\frac{1}{2}$-power ($\Delta = 0$) instead of $\frac{1}{2+\Delta}$ with $\Delta > 0$, $b_t$ will grow faster compared with the same term in AdaGradNorm. We will see later that $\Delta = 0$ still leads to the convergence of the last iterate. However, we first focus on the easier case with $\Delta > 0$ and state convergence rate of Algorithm 3 in Theorem 4.1.

**Theorem 4.1.** *With Assumptions 1 and 2', by taking $p_t = \frac{1}{t}$ in Algorithm 3, we have*

$$F(x_{T+1}) - F^* \leq \frac{\left(\frac{2}{\eta}\left(\frac{\|x_1-x^*\|^2}{\gamma\eta} + h(\Delta) + g(\Delta)\right) + b_0^\Delta\right)^{\frac{1}{\Delta}}\left(\frac{\|x_1-x^*\|^2}{\gamma\eta} + h(\Delta) + g(\Delta)\right)}{T}$$

*where*

$$h(\Delta) := \begin{cases} \frac{(2+\Delta)\eta(\eta L)^\Delta}{2}\log^+ \frac{\eta L}{b_0} & \Delta \geq 1 \\ \frac{(2+\Delta)\eta^2 L}{2b_0^{1-\Delta}}\log^+ \frac{\eta L}{b_0} & \Delta \in (0,1) \end{cases} \quad and \quad g(\Delta) := \frac{(2+\Delta)\eta}{\gamma}\left(\frac{2\eta L}{\gamma}\right)^\Delta \log^+ \frac{2\eta L}{\gamma b_0}.$$

An issue with variant 3 is that, when using $\frac{1}{2+\Delta}$-power, the stepsize ceases to be scale-invariant. Algorithm 4 shows a different approach, using the scale-invariant power $\frac{1}{2}$, but a different stepsize $b_t^\delta b_{t-1}^{1-\delta}$, for a constant $\delta \in [2/3, 1)$. The tradeoff is that the provable convergence rate of the second variant depends exponentially on the smoothness parameter. We also note that, when $\delta = 1$, we obtain the same algorithm as when setting $\Delta = 0$ in the previous variant.

*Remark* 4.2. $b_0$ in every algorithm is only for stabilization and is set to a constant that is very close to 0 in practice. However, the first stepsize in Algorithm 4, i.e., $b_1^\delta b_0^{1-\delta}$ will explode. To avoid this issue, we can simply set the first stepsize as $b_1$ instead of $b_1^\delta b_0^{1-\delta}$. We note that, under this change, Algorithm 4 still admits a provable convergence rate. However, for simplicity, we keep $b_1^\delta b_0^{1-\delta}$ in both the description of the algorithm and its analysis.

**Theorem 4.3.** *With Assumptions 1 and 2', by taking $p_t = \frac{1}{t}$ in Algorithm 4, we have*

$$F(x_{T+1}) - F^* \leq \frac{b_0 \exp\left(\frac{k(\delta)}{1-\delta}\right)k(\delta)}{T},$$

*where* $k(\delta) = \frac{\|x_1 - x^*\|^2}{\gamma \eta^2} + \frac{\eta L}{b_0} \left(1 - \left(\frac{b_0}{\eta L}\right)^{\frac{1}{\delta}}\right)^{+} + \frac{2}{\gamma \delta} \left(\frac{2\eta L}{\gamma b_0}\right)^{\frac{2}{\delta} - 2} \log^{+} \frac{2\eta L}{\gamma b_0}.$

To finish this section, we briefly discuss the case when $\Delta = 0$ in Algorithm 3 or equivalently $\delta = 1$ in Algorithm 4. First, by seeing $\Delta$ tends to 0, we can expect a convergence rate depending exponentially on the problem parameters. When $\Delta = 0$, while we can still expect a bound of the function gap via the final stepsize $b_T$, bounding $b_T$ becomes problematic. In the proof of Theorem 4.1, to bound $b_T$, we use the sum $\sum_{t=1}^{T} \frac{\|\nabla F(x_t)\|^2}{b_t^2 p_t} = \sum_{t=1}^{T} \frac{b_t^{2+\Delta} - b_{t-1}^{2+\Delta}}{b_t^2}$. This sum only admits a lower bound in terms of $b_T$ when $\Delta > 0$, thus the argument does not work when $\Delta = 0$. However, it is still possible to give an asymptotic rate under the $\gamma$-quasar convexity assumption. If we further assume that $F$ is convex, we can give a non-asymptotic rate. The main idea on how to bound $b_T$ is as follows. Let $\tau$ be the last time such that $b_t \leq \eta L / 2$. The increment from $b_{\tau+1}$ to $b_T$ can be bounded by observing that the increase in each step $\|\nabla F(x_t)\|^2 \leq \frac{2}{3} p_t b_t^2$. Moreover, the critical step is the increase from $b_\tau$ to $b_{\tau+1}$, which again can be analyzed via the function gap and smoothness. We present the asymptotic and a non-symptotic convergence rate and their analysis in Sections B.4 and B.5 in the appendix.

## 5 ACCELERATED VARIANTS OF ADAGRADNORM FOR CONVEX AND SMOOTH MINIMIZATION ON $\mathbb{R}^d$

In this section, by using the stronger Assumption 1', we give two new algorithms that achieve the accelerated rate $O(1/T^2)$, matching the optimal rate in $T$ for convex and smooth optimization for unconstrained deterministic problems. Our new algorithms are adapted from the acceleration scheme introduced in Auslender & Teboulle (2006) (see also Lan (2020)). They are also similar to existing adaptive accelerated methods designed for bounded domains, including Levy et al. (2018); Ene et al. (2021). However, previous analysis does not apply in unconstrained problems; we therefore have to make necessary modifications.

To the best of our knowledge, in unconstrained problems under the deterministic setting, the only existing analysis for an accelerated method was introduced in Antonakopoulos et al. (2022). Here we discuss some limitations of this work. The convergence rate for the weighted average iterate $\overline{x}_{T+1/2}$ is given by

$$f(\overline{x}_{T+1/2}) - f(x) \leq O\left(\frac{1}{T^2}\left(R_h \lim_{t \to \infty} b_t + K_h \lim_{t \to \infty} b_t^2\right)\right)$$

where $h$ is a $K_h$-strongly convex mirror map function, $R_h = \max h(x) - \min h(x)$ is the range of $h$. This result is only applicable when the domain is unbounded but the range of the mirror map is bounded. Even in the standard $\ell_2$ setup with $h(x) = \frac{1}{2}\|x\|^2$, this assumption does not hold. Moreover, due to the term $\lim_{t \to \infty} b_t$, the above guarantee is dependent on the particular function. Thus, while a standard convergence guarantee is applicable to say, all SVM models with Huber loss, the above guarantee varies for each SVM model and there is no universal bound for all of them.

We further highlight some key differences between this work and ours. While the convergence rate above depends on the convergence of the stepsize, for both our variants, we will show an explicit convergence rate that holds universally for the entire function class. Second, the algorithm in Antonakopoulos et al. (2022) is based on an extra gradient method which requires to calculate gradients twice in one iteration. Instead, our algorithms only need one gradient computation per iteration. Finally, our algorithms guarantee the convergence of the last iterate as opposed to that for the weighted average iterate as shown above.

Algorithm 5 shows the first variant. For an accelerated method, the step size typically has the form $b_t = \left(b_0^2 + \sum_{i=1}^{t} s_i \|\nabla F_i\|^2\right)^{\frac{1}{2}}$ where $\nabla F_i$ is the gradient evaluated at time $i$, and $s_i = O(i^2)$. However in order to be able to give an explicit convergence rate, Algorithm 5 uses a smaller $b_t$ with power $\frac{1}{2+\Delta}$, with $\Delta > 0$. When $\Delta = 0$, we can only show an asymptotic convergence rate, similarly to Antonakopoulos et al. (2022). We first focus on the case when $\Delta > 0$. In the appendix we will discuss the convergence of the algorithm when $\Delta = 0$. We have the following theorem.

| **Algorithm 5** AdaGradNorm-Acc | **Algorithm 6** AdaGradNorm-Acc |
|---|---|
| Initialize: $x_1 = w_1, \eta > 0, \Delta > 0, a_t > 0, q_t > 0$ | Initialize: $x_1 = w_1, \eta > 0, \delta \in [2/3, 1), a_t > 0, q_t > 0$ |
| for $t = 1$ to $T$ | for $t = 1$ to $T$ |
| $\quad v_t = (1 - a_t)w_t + a_t x_t$ | $\quad v_t = (1 - a_t)w_t + a_t x_t$ |
| $\quad b_t = \left(b_0^{2+\Delta} + \sum_{i=1}^t \frac{\|\nabla F(v_i)\|^2}{q_i^2}\right)^{\frac{1}{2+\Delta}}$ | $\quad b_t = \left(b_0^2 + \sum_{i=1}^t \frac{\|\nabla F(v_i)\|^2}{q_i^2}\right)^{\frac{1}{2}}$ |
| $\quad x_{t+1} = x_t - \frac{\eta}{q_t b_t}\nabla F(v_t)$ | $\quad x_{t+1} = x_t - \frac{\eta}{q_t b_t^\delta b_{t-1}^{1-\delta}}\nabla F(v_t)$ |
| $\quad w_{t+1} = (1 - a_t)w_t + a_t x_{t+1}$ | $\quad w_{t+1} = (1 - a_t)w_t + a_t x_{t+1}$ |

**Theorem 5.1.** *Suppose $F$ satisfies Assumptions 1' and 2', let $a_t = \frac{2}{t+1}$, $q_t = \frac{2}{t}$ in Alg. 5, then*

$$F(w_{T+1}) - F^* \leq \frac{1}{T(T+1)}\left(\frac{2\|x^* - x_1\|^2}{\eta^2} + \frac{4h(\Delta)}{\eta} + b_0^\Delta\right)^{\frac{1}{\Delta}}\left(\frac{\|x^* - x_1\|^2}{2\eta} + h(\Delta)\right)$$

*where*

$$h(\Delta) = \begin{cases} \frac{(2+\Delta)(2\eta L)^{\Delta-1}L\eta^2}{2}\log^+ \frac{2\eta L}{b_0} & \Delta \geq 1 \\ \frac{(2+\Delta)L\eta^2}{2b_0^{1-\Delta}}\log^+ \frac{2\eta L}{b_0} & \Delta \in (0, 1) \end{cases}.$$

Similarly to the second variant in the previous section, we also have a scale-invariant accelerated algorithm, shown in Algorithm 6 using power $\frac{1}{2}$ but a smaller stepsize $b_t^\delta b_{t-1}^{1-\delta}$. This algorithm also has an exponential dependency on the problem parameters, which is given in the following theorem.

*Remark* 5.2. Similar to Remark 4.2, the first stepsize in Algorithm 6, i.e., $b_1^\delta b_0^{1-\delta}$ can be replaced by $b_1$. However, for simplicity, we keep $b_1^\delta b_0^{1-\delta}$ in both the description of the algorithm and its analysis.

**Theorem 5.3.** *Suppose $F$ satisfies Assumptions 1' and 2', let $a_t = \frac{2}{t+1}$, $q_t = \frac{2}{t}$ in Alg. 6, then*

$$F(w_{T+1}) - F^* \leq \frac{b_0 \exp\left(\frac{s(\delta)}{1-\delta}\right)(s(\delta))}{T(T+1)},$$

*where* $s(\delta) = \frac{\|x^* - x_1\|^2}{2\eta} + \frac{\eta^2 L}{b_0}\left(1 - \left(\frac{b_0}{2\eta L}\right)^{\frac{1}{\delta}}\right)^+.$

Similarly to the previous section, we give a more detailed discussion of the convergence of the Algorithm 5 when $\Delta = 0$ or equivalently Algorithm 6 when $\delta = 1$ in Section C.4 in the appendix. While we can still show an accelerated $O(1/T^2)$ asymptotic convergence rate, we only present an $O(1/T^2 + 1/T)$ non-asymptotic rate. The difference between these algorithms and the ones in the previous section is that the stepsize $b_t$ increases much faster. More precisely, the increment in each step is now $O(t^2\|\nabla F(v_t)\|^2)$ instead of $O(t\|\nabla F(x_t)\|^2)$. Thus we can only show an upperbound for $b_t$ that grows linearly with time, which leads to the $O(1/T^2 + 1/T)$ convergence rate.

## 6 CONCLUSION AND FUTURE WORK

In this paper, we go back to the most basic AdaGrad algorithm and study its convergence rate in generalized smooth convex optimization. We prove explicit convergence guarantees for unconstrained problems in both the deterministic and stochastic setting. Building on these insights, we propose new algorithms that exhibit last iterate convergence, with and without acceleration. We see our work as primarily theoretical since the first and foremost goal is to understand properties of existing algorithms that work well in practice. We refer the reader to the long line of previous works (Duchi et al., 2011; Levy, 2017; Kavis et al., 2019; Bach & Levy, 2019; Antonakopoulos et al., 2020; Ene et al., 2021; Ene & Nguyen, 2022; Antonakopoulos et al., 2022) that have already demonstrated the behavior of AdaGrad and accelerated adaptive algorithms empirically.

## ACKNOWLEDGMENTS

TN and AE were supported in part by NSF CAREER grant CCF-1750333, NSF grant III-1908510, and an Alfred P. Sloan Research Fellowship. HN was supported in part by NSF CAREER grant CCF-1750716 and NSF grant CCF-1909314.

**Reproducibility Statement.** We include the full proofs of all theorems in the Appendix.

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

# A  MISSING PROOFS FROM SECTION 3

## A.1  ADAGRADNORM

As we pointed out before, it is possible to obtain an improvement by a factor $1/\gamma$ compared with Theorem 3.1 by assuming $L$-smoothness instead of weak $L$-smoothness.

**Theorem A.1.** *With Assumptions 1 and 2', AdaGradNorm admits*

$$\frac{\sum_{t=1}^{T} F(x_t) - F^*}{T} \leq \frac{\left( \frac{L\|x_1 - x^*\|^2}{\eta} + 2\eta L \log^+ \frac{\eta L}{b_0} + b_0 \right) \left( \frac{\|x_1 - x^*\|^2}{\gamma\eta} + \frac{2\eta}{\gamma} \log^+ \frac{2\eta L}{b_0} \right)}{T}.$$

*Proof.* Note that Assumption 2' can imply Assumption 2, so following the same proof of Theorem 3.1, we still have

$$\sum_{t=1}^{T} F(x_t) - F^* \leq b_T \left( \frac{\|x_1 - x^*\|^2}{\gamma\eta} + \frac{2\eta}{\gamma} \log^+ \frac{2\eta L}{\gamma b_0} \right).$$

However, from here, we will bound $b_T$ directly, rathe than use the self bounded argument in the previous proof. By the $L$-smoothness, we know

$$F(x_{t+1}) - F(x_t) \leq \langle \nabla F(x_t), x_{t+1} - x_t \rangle + \frac{L}{2} \|x_{t+1} - x_t\|^2$$

$$= \left( \frac{L\eta^2}{2b_t^2} - \frac{\eta}{b_t} \right) \|\nabla F(x_t)\|^2$$

$$\Rightarrow \frac{\|\nabla F(x_t)\|^2}{b_t} \leq \frac{2\left(F(x_t) - F(x_{t+1})\right)}{\eta} + \left( \frac{L\eta}{b_t^2} - \frac{1}{b_t} \right) \|\nabla F(x_t)\|^2.$$

Sum up from 1 to $T$, we know

$$\sum_{t=1}^{T} \frac{\|\nabla F(x_t)\|^2}{b_t} \leq \frac{2}{\eta} \left( F(x_1) - F(x_{T+1}) \right) + \sum_{t=1}^{T} \left( \frac{L\eta}{b_t^2} - \frac{1}{b_t} \right) \|\nabla F(x_t)\|^2$$

$$\leq \frac{2}{\eta} \left( F(x_1) - F(x^*) \right) + \sum_{t=1}^{T} \left( \frac{L\eta}{b_t^2} - \frac{1}{b_t} \right) \|\nabla F(x_t)\|^2.$$

Use the the same proof technique as before, we can bound

$$\sum_{t=1}^{T} \left( \frac{L\eta}{b_t^2} - \frac{1}{b_t} \right) \|\nabla F(x_t)\|^2 \leq 2\eta L \log^+ \frac{\eta L}{b_0}.$$

and

$$\sum_{t=1}^{T} \frac{\|\nabla F(x_t)\|^2}{b_t} = \sum_{t=1}^{T} \frac{b_t^2 - b_{t-1}^2}{b_t} \geq \sum_{t=1}^{T} b_t - b_{t-1} = b_T - b_0.$$

Hence, we know

$$b_T \leq \frac{2}{\eta} \left( F(x_1) - F(x^*) \right) + 2\eta L \log^+ \frac{\eta L}{b_0} + b_0$$

$$\leq \frac{L\|x_1 - x^*\|^2}{\eta} + 2\eta L \log^+ \frac{\eta L}{b_0} + b_0.$$

By using this bound on $b_T$, we can get the final result with an improvement by a factor $1/\gamma$.  □

## A.2 Stochastic AdaGradNorm

We will employ the following notations for convenience

$$
\Delta_t := \sum_{s=1}^{t} F(x_s) - F^*;
$$

$$
\xi_t := \widehat{\nabla} F(x_t) - \nabla F(x_t);
$$

$$
M_t := \max_{s \in [t]} \|\xi_s\|^2.
$$

Before diving into the details of our proof, we first present some technical results we will use in the proof of Theorem 3.5.

### A.2.1 Technical lemmas

To start with, under Assumptions 1 and 3 only, we can obtain a bound for a term close to our final goal $\Delta_T$.

**Lemma A.2.** *(Lemma 3.4) Suppose F satisfies Assumptions 1 and 3, we have*

$$
\mathbb{E}\left[\frac{\Delta_T}{b_T}\right] \leq \frac{\|x_1 - x^*\|^2}{\gamma\eta} + \frac{2\eta}{\gamma} \mathbb{E}\left[\frac{M_T}{b_0^2} + \log\frac{b_T}{b_0}\right].
$$

*Proof.* We start by using the $\gamma$-quasar convexity of the function $F$

$$
\begin{aligned}
F(x_t) - F^* &\leq \frac{\langle \nabla F(x_t), x_t - x^* \rangle}{\gamma} \\
&= \frac{\langle \nabla F(x_t) - \widehat{\nabla} F(x_t), x_t - x^* \rangle}{\gamma} + \frac{\langle \widehat{\nabla} F(x_t), x_t - x^* \rangle}{\gamma} \\
&= \frac{\langle -\xi_t, x_t - x^* \rangle}{\gamma} + \frac{b_t}{2\gamma\eta}\left(\|x_t - x^*\|^2 - \|x_{t+1} - x^*\|^2 + \|x_{t+1} - x_t\|^2\right).
\end{aligned}
$$

Dividing both sides by $2b_t - b_0$ and taking expactations, we have

$$
\mathbb{E}\left[\frac{F(x_t) - F^*}{2b_t - b_0}\right] \leq \mathbb{E}\left[\frac{\langle -\xi_t, x_t - x^* \rangle}{\gamma(2b_t - b_0)} + \frac{b_t}{2b_t - b_0} \times \frac{\|x_t - x^*\|^2 - \|x_{t+1} - x^*\|^2 + \|x_{t+1} - x_t\|^2}{2\gamma\eta}\right].
$$

Now we no longer have a telescoping sum in the R.H.S.. However, this is exactly what we want to handle the difficult term $\frac{\langle -\xi_t, x_t - x^* \rangle}{\gamma(2b_t - b_0)}$ which does not disappear after taking the expectation. The key step is to use Cauchy-Schwarz inequality for the term

$$
|\langle -\xi_t, x_t - x^* \rangle| \leq \frac{\lambda}{2} \|\xi_t\|^2 + \frac{1}{2\lambda} \|x_t - x^*\|^2
$$

with the appropriate coefficient $\lambda$ so that the term $\|x_t - x^*\|^2$ can be absorbed to make a telescoping sum $\frac{b_{t-1}\|x_t - x^*\|^2}{2b_{t-1} - b_0} - \frac{b_t\|x_{t+1} - x^*\|^2}{2b_t - b_0}$. The remaining terms are free of $x^*$; hence can be more easily

bounded. To do this, note that

$$
\mathbb{E}\left[\frac{\langle -\xi_t, x_t - x^* \rangle}{\gamma(2b_t - b_0)}\right]
$$

$$
=\mathbb{E}\left[\underbrace{\frac{1}{\gamma}\left(\frac{1}{2b_t - b_0} - \frac{1}{2b_{t-1} - b_0}\right)}_{A}\langle -\xi_t, x_t - x^* \rangle\right]
$$

$$
\leq\mathbb{E}\left[|A|\,|\langle -\xi_t, x_t - x^* \rangle|\right]
$$

$$
\leq\mathbb{E}\left[|A|\left(\frac{|A|}{4}\left(\frac{b_{t-1}}{2\gamma\eta(2b_{t-1} - b_0)} - \frac{b_t}{2\gamma\eta(2b_t - b_0)}\right)^{-1}\|\xi_t\|^2\right.\right.
$$

$$
\left.\left.+|A|^{-1}\left(\frac{b_{t-1}}{2\gamma\eta(2b_{t-1} - b_0)} - \frac{b_t}{2\gamma\eta(2b_t - b_0)}\right)\|x_t - x^*\|^2\right)\right]
$$

$$
=\mathbb{E}\left[\frac{\eta}{\gamma b_0}\left(\frac{1}{2b_{t-1} - b_0} - \frac{1}{2b_t - b_0}\right)\|\xi_t\|^2\right]
$$

$$
+\mathbb{E}\left[\left(\frac{b_{t-1}}{2b_{t-1} - b_0} - \frac{b_t}{2b_t - b_0}\right)\frac{\|x_t - x^*\|^2}{2\gamma\eta}\right]
$$

Thus we have

$$
\mathbb{E}\left[\frac{F(x_t) - F^*}{2b_t - b_0}\right]
$$

$$
\leq\mathbb{E}\left[\frac{\eta}{\gamma b_0}\left(\frac{1}{2b_{t-1} - b_0} - \frac{1}{2b_t - b_0}\right)\|\xi_t\|^2\right]
$$

$$
+\mathbb{E}\left[\frac{b_{t-1}\|x_t - x^*\|^2}{2\gamma\eta(2b_{t-1} - b_0)} - \frac{b_t\|x_{t+1} - x^*\|^2}{2\gamma\eta(2b_t - b_0)}\right] + \mathbb{E}\left[\frac{\eta\left\|\widehat{\nabla}F(x_t)\right\|^2}{2\gamma b_t(2b_t - b_0)}\right]
$$

$$
\leq\mathbb{E}\left[\frac{\eta}{\gamma b_0}\left(\frac{1}{2b_{t-1} - b_0} - \frac{1}{2b_t - b_0}\right)\|\xi_t\|^2\right]
$$

$$
+\mathbb{E}\left[\frac{b_{t-1}\|x_t - x^*\|^2}{2\gamma\eta(2b_{t-1} - b_0)} - \frac{b_t\|x_{t+1} - x^*\|^2}{2\gamma\eta(2b_t - b_0)}\right] + \mathbb{E}\left[\frac{\eta\left\|\widehat{\nabla}F(x_t)\right\|^2}{2\gamma b_t^2}\right]
$$

Now we have a telescoping sum

$$
\frac{b_{t-1}\|x_t - x^*\|^2}{2\gamma\eta(2b_{t-1} - b_0)} - \frac{b_t\|x_{t+1} - x^*\|^2}{2\gamma\eta(2b_t - b_0)}
$$

and the remaining terms are free of $x_t - x^*$. Taking the sum over $t$, we have

$$
\mathbb{E}\left[\sum_{t=1}^{T}\frac{F(x_t) - F^*}{2b_t - b_0}\right]
$$

$$
\leq\mathbb{E}\left[\sum_{t=1}^{T}\frac{\eta}{2\gamma b_0}\left(\frac{1}{2b_t - b_0} - \frac{1}{2b_{t-1} - b_0}\right)\|\xi_t\|^2\right]
$$

$$
+\frac{\|x_1 - x^*\|^2}{2\gamma\eta} + \mathbb{E}\left[\sum_{t=1}^{T}\frac{\eta\|\widehat{\nabla}F(x_t)\|^2}{2\gamma b_t^2}\right]. \tag{2}
$$

First for the easy term $\mathbb{E}\left[\sum_{t=1}^{T} \frac{\eta\|\widehat{\nabla}F(x_t)\|^2}{2\gamma b_t^2}\right]$, we have

$$
\begin{aligned}
\mathbb{E}\left[\sum_{t=1}^{T} \frac{\eta\|\widehat{\nabla}F(x_t)\|^2}{2\gamma b_t^2}\right] &= \frac{\eta}{2\gamma}\mathbb{E}\left[\sum_{t=1}^{T}\frac{b_t^2 - b_{t-1}^2}{b_t^2}\right] \\
&\leq \frac{\eta}{2\gamma}\mathbb{E}\left[\sum_{t=1}^{T}\log b_t^2 - \log b_{t-1}^2\right] \\
&= \frac{\eta}{\gamma}\mathbb{E}\left[\log\frac{b_T}{b_0}\right].
\end{aligned}
\tag{3}
$$

Next, we bound

$$
\begin{aligned}
&\mathbb{E}\left[\sum_{t=1}^{T}\frac{\eta}{\gamma b_0}\left(\frac{1}{2b_{t-1}-b_0} - \frac{1}{2b_t - b_0}\right)\|\xi_t\|^2\right] \\
\leq &\mathbb{E}\left[\sum_{t=1}^{T}\frac{\eta}{\gamma b_0}\left(\frac{1}{2b_{t-1}-b_0} - \frac{1}{2b_t - b_0}\right)M_T\right] \\
\leq &\mathbb{E}\left[\frac{\eta}{\gamma b_0^2}M_T\right]
\end{aligned}
\tag{4}
$$

Plugging the bounds (3) and (4) into (2), we have

$$
\mathbb{E}\left[\sum_{t=1}^{T}\frac{F(x_t)-F^*}{2b_t}\right] \leq \mathbb{E}\left[\sum_{t=1}^{T}\frac{F(x_t)-F^*}{2b_t - b_0}\right] \leq \frac{\|x_1 - x^*\|^2}{2\gamma\eta} + \frac{\eta}{\gamma b_0^2}\mathbb{E}[M_T] + \frac{\eta}{\gamma}\mathbb{E}\left[\log\frac{b_T}{b_0}\right].
$$

The last step is using $\sum_{t=1}^{T}\frac{F(x_t)-F^*}{2b_t} \geq \frac{\sum_{t=1}^{T}F(x_t)-F^*}{2b_T} = \frac{\Delta_T}{2b_T}$ to finish the proof. $\qquad\square$

Due to the appearance of $M_T$ in Lemma A.2, it is natural to consider what we can obtain under the additional Assumption 4, i.e., sub-Weibull noise with parameter $\theta$. We first provide the following simple bound on $\mathbb{E}\left[\|\xi_t\|^2\right]$. The result is not new and the proof is only included for completeness.

**Lemma A.3.** *Under Assumption 4, $\forall t \in [T]$, we have*

$$
\mathbb{E}\left[\|\xi_t\|^2\right] \leq \Gamma(2\theta + 1)e\sigma^2.
$$

*Proof.* We first note that from the definition of sub-Weibull noise, the tail of $\|\xi_t\|$ can be bounded as follows

$$
\Pr\left[\|\xi_t\| \geq u\right] \leq \frac{\mathbb{E}\left[\exp\left((\|\xi_t\|/\sigma)^{1/\theta}\right)\right]}{\exp\left((u/\sigma)^{1/\theta}\right)} \leq \exp\left(1 - (u/\sigma)^{1/\theta}\right).
$$

Then we can obtain

$$
\begin{aligned}
\mathbb{E}\left[\|\xi_t\|^2\right] &= \int_0^{\infty} 2u\Pr\left[\|\xi\| \geq u\right]\mathrm{d}u \\
&\leq \int_0^{\infty} 2u\exp\left(1 - (u/\sigma)^{1/\theta}\right)\mathrm{d}u \\
&= 2\theta e\sigma^2 \int_0^{\infty} v^{2\theta - 1}\exp(-v)\mathrm{d}v \\
&= \Gamma(2\theta + 1)e\sigma^2
\end{aligned}
$$

where $u$ is substituted by $\sigma v^\theta$ in the second equation. $\qquad\square$

Next, we prove a high probability bound on $M_T$, the proof of which is inspired by Lemma 5 in Li & Orabona (2020).

**Lemma A.4.** *Under Assumption 4, given $0 < \delta < 1$, define the event*

$$E(\delta) = \left\{ M_T \le \sigma^2 \log^{2\theta} \frac{eT}{\delta} \right\},$$

*we have $\Pr\left[E(\delta)\right] \ge 1 - \delta$.*

*Proof.* Note that

$$
\begin{aligned}
\Pr\left[M_T \ge u\right] &= \Pr\left[\max_{s \in [T]} \|\xi_s\|^2 \ge u\right] \\
&= \Pr\left[\max_{s \in [T]} \|\xi_s\|^{\frac{1}{\theta}} \ge u^{\frac{1}{2\theta}}\right] \\
&\le \frac{\mathbb{E}\left[\exp\left(\max_{s \in [T]}(\|\xi_s\|/\sigma)^{1/\theta}\right)\right]}{\exp\left((u^{1/2}/\sigma)^{1/\theta}\right)} \\
&\le \frac{\sum_{s=1}^{T} \mathbb{E}\left[\exp\left((\|\xi_s\|/\sigma)^{1/\theta}\right)\right]}{\exp\left((u^{1/2}/\sigma)^{1/\theta}\right)} \\
&= T \exp\left(1 - (u^{1/2}/\sigma)^{1/\theta}\right).
\end{aligned}
$$

Choose $u = \sigma^2 \log^{2\theta} \frac{eT}{\delta}$ to obtain

$$\Pr\left[M_T \ge \sigma^2 \log^{2\theta} \frac{eT}{\delta}\right] \le \delta.$$

$\square$

Lastly, we will find an upper bound on the $p$-th moment of $M_T$.

**Lemma A.5.** *Under Assumption 4, given $p > 0$, there is*

$$\mathbb{E}\left[M_T^p\right] \le \sigma^{2p}\left(\log^{2\theta p}\left(\Gamma(4\theta p + 1)e^2 T^2\right) + 1\right).$$

*Proof.* Note that in Lemma A.4, we proved

$$\Pr\left[M_T \ge u\right] \le T \exp\left(1 - (u^{1/2}/\sigma)^{1/\theta}\right).$$

Let $E(\delta)$ be the same as it in Lemma A.4. Then, by Holder's inequality we have

$$
\begin{aligned}
\mathbb{E}\left[M_T^p\right] &= \mathbb{E}\left[M_T^p \mathbb{1}_{E(\delta)}\right] + \mathbb{E}\left[M_T^p \mathbb{1}_{E^c(\delta)}\right] \\
&\le \mathbb{E}\left[M_T^p \mathbb{1}_{E(\delta)}\right] + \sqrt{\mathbb{E}\left[M_T^{2p}\right]\mathbb{E}\left[\mathbb{1}_{E^c(\delta)}\right]} \\
&\le \sigma^{2p}\log^{2\theta p}\frac{eT}{\delta} + \sqrt{\mathbb{E}\left[M_T^{2p}\right]\delta} \\
&= \sigma^{2p}\log^{2\theta p}\frac{eT}{\delta} + \sqrt{\delta \int_0^\infty 2pu^{2p-1}\Pr\left[M_T \ge u\right]\mathrm{d}u} \\
&\le \sigma^{2p}\log^{2\theta p}\frac{eT}{\delta} + \sqrt{\delta \int_0^\infty 2pu^{2p-1}T\exp\left(1 - (u^{1/2}/\sigma)^{1/\theta}\right)\mathrm{d}u} \\
&= \sigma^{2p}\log^{2\theta p}\frac{eT}{\delta} + \sigma^{2p}\sqrt{\Gamma(4\theta p + 1)eT\delta} \\
&= \sigma^{2p}\left(\log^{2\theta p}\frac{eT}{\delta} + \sqrt{\Gamma(4\theta p + 1)eT\delta}\right).
\end{aligned}
$$

Choose $\delta = \frac{1}{\Gamma(4\theta p + 1)eT} < 1$, we have

$$\mathbb{E}\left[M_T^p\right] \le \sigma^{2p}\left(\log^{2\theta p}\left(\Gamma(4\theta p + 1)e^2 T^2\right) + 1\right).$$

$\square$

Note that all the above results only depend on Assumptions 1, 3 and 4 without requiring the smoothness of $F$.

### A.2.2 PROOF OF THEOREM 3.5

Theorem 3.5 requires Assumption 2' additionally. Thus we first show that under Assumptions 2' and 4. $b_T$ enjoys a $\widetilde{O}(1 + \sigma\sqrt{T \log^{2\theta} T})$ upper bound with high probability.

**Lemma A.6.** *(Lemma 3.3) Suppose $F$ satisfies Assumptions 2' and 4. Under the event $E(\delta) = \left\{ M_T \le \sigma^2 \log^{2\theta} \frac{eT}{\delta} \right\}$, we have*

$$b_T \le g_T(\delta) := 2b_0 + \frac{4(F(x_1) - F^*)}{\eta} + 4\eta L \log^+ \frac{\eta L}{b_0} + 4\sigma \sqrt{T \log^{2\theta} \frac{eT}{\delta} \log\left(1 + \frac{16\sigma^2 T \log^{2\theta} \frac{eT}{\delta}}{b_0^2}\right)}.$$

*Additionally, by Lemma A.4, there is*

$$1 - \delta \le \Pr\left[E(\delta)\right] \le \Pr\left[b_T \le g_T(\delta)\right].$$

*Proof.* We start by using the smoothness of $F$

$$\begin{aligned}
F(x_{t+1}) - F(x_t) &\le \langle \nabla F(x_t), x_{t+1} - x_t \rangle + \frac{L}{2}\|x_{t+1} - x_t\|^2 \\
&= -\frac{\eta}{b_t}\langle \nabla F(x_t), \widehat{\nabla}F(x_t)\rangle + \frac{\eta^2 L}{2b_t^2}\|\widehat{\nabla}F(x_t)\|^2 \\
&= -\frac{\eta}{b_t}\langle \nabla F(x_t) - \widehat{\nabla}F(x_t), \widehat{\nabla}F(x_t)\rangle - \frac{\eta}{b_t}\|\widehat{\nabla}F(x_t)\|^2 + \frac{\eta^2 L}{2b_t^2}\|\widehat{\nabla}F(x_t)\|^2 \\
\Rightarrow \frac{\|\widehat{\nabla}F(x_t)\|^2}{b_t} &\le \frac{2}{\eta}(F(x_t) - F(x_{t+1})) + \frac{2\langle \xi_t, \widehat{\nabla}F(x_t)\rangle}{b_t} + \left(\frac{\eta L}{b_t^2} - \frac{1}{b_t}\right)\|\widehat{\nabla}F(x_t)\|^2.
\end{aligned}$$

Taking the sum over $t$ we have

$$\sum_{t=1}^{T} \frac{\|\widehat{\nabla}F(x_t)\|^2}{b_t} \le \frac{2(F(x_1) - F^*)}{\eta} + 2\sum_{t=1}^{T} \frac{\langle \xi_t, \widehat{\nabla}F(x_t)\rangle}{b_t} + \sum_{t=1}^{T}\left(\frac{\eta L}{b_t^2} - \frac{1}{b_t}\right)\|\widehat{\nabla}F(x_t)\|^2.$$

Using the common technique, we know that $\sum_{t=1}^{T}\left(\frac{\eta L}{b_t^2} - \frac{1}{b_t}\right)\|\widehat{\nabla}F(x_t)\|^2 \le 2\eta L \log^+ \frac{\eta L}{b_0}$. Moreover, for the L.H.S.

$$\sum_{t=1}^{T} \frac{\|\widehat{\nabla}F(x_t)\|^2}{b_t} = \sum_{t=1}^{T} \frac{b_t^2 - b_{t-1}^2}{b_t} \ge \sum_{t=1}^{T} b_t - b_{t-1} = b_T - b_0.$$

Thus we have

$$b_T \le b_0 + \frac{2(F(x_1) - F^*)}{\eta} + 2\eta L \log^+ \frac{\eta L}{b_0} + 2\sum_{t=1}^{T} \frac{\langle \xi_t, \widehat{\nabla}F(x_t)\rangle}{b_t}$$

For the last term in this equation, we notice that $\langle \xi_t, \widehat{\nabla} F(x_t) \rangle \leq \|\xi_t\| \|\widehat{\nabla} F(x_t)\| \leq \sqrt{M_T} \|\widehat{\nabla} F(x_t)\|$, hence

$$b_T \leq \frac{2(F(x_1) - F^*)}{\eta} + 2\eta L \log^+ \frac{\eta L}{b_0} + b_0 + 2 \sum_{t=1}^{T} \frac{\langle \xi_t, \widehat{\nabla} F(x_t) \rangle}{b_t}$$

$$\leq \frac{2(F(x_1) - F^*)}{\eta} + 2\eta L \log^+ \frac{\eta L}{b_0} + b_0 + 2\sqrt{M_T} \sum_{t=1}^{T} \frac{\|\widehat{\nabla} F(x_t)\|}{b_t}$$

$$\overset{(a)}{\leq} \frac{2(F(x_1) - F^*)}{\eta} + 2\eta L \log^+ \frac{\eta L}{b_0} + b_0 + 2\sqrt{M_T} \sqrt{T \sum_{t=1}^{T} \frac{\|\widehat{\nabla} F(x_t)\|^2}{b_t^2}}$$

$$= \frac{2(F(x_1) - F^*)}{\eta} + 2\eta L \log^+ \frac{\eta L}{b_0} + b_0 + \sqrt{4 M_T T \sum_{t=1}^{T} \frac{b_t^2 - b_{t-1}^2}{b_t^2}}$$

$$\leq \frac{2(F(x_1) - F^*)}{\eta} + 2\eta L \log^+ \frac{\eta L}{b_0} + b_0 + \sqrt{4 M_T T \log \frac{b_T^2}{b_0^2}}$$

where $(a)$ is due to Jensen's inequality. We can write

$$4 M_T T \log \frac{b_T^2}{b_0^2} = 4 M_T T \left( \log \frac{b_T^2}{b_0^2 + 16 M_T T} + \log \frac{b_0^2 + 16 M_T T}{b_0^2} \right)$$

$$\leq 4 M_T T \left( \frac{b_T^2}{b_0^2 + 16 M_T T} + \log \frac{b_0^2 + 16 M_T T}{b_0^2} \right)$$

$$\leq \frac{b_T^2}{4} + 4 M_T T \log \frac{b_0^2 + 16 M_T T}{b_0^2}.$$

Hence

$$b_T \leq b_0 + \frac{2(F(x_1) - F^*)}{\eta} + 2\eta L \log^+ \frac{\eta L}{b_0} + \sqrt{\frac{b_T^2}{4} + 4 M_T T \log \frac{b_0^2 + 16 M_T T}{b_0^2}}$$

$$\leq b_0 + \frac{2(F(x_1) - F^*)}{\eta} + 2\eta L \log^+ \frac{\eta L}{b_0} + \frac{b_T}{2} + 2\sqrt{M_T T \log \frac{b_0^2 + 16 M_T T}{b_0^2}}$$

which gives us

$$b_T \leq 2 b_0 + \frac{4(F(x_1) - F^*)}{\eta} + 4\eta L \log^+ \frac{\eta L}{b_0} + 4\sqrt{M_T T \log \frac{b_0^2 + 16 M_T T}{b_0^2}}.$$

Recall the definition of the event $E(\delta)$ is $M_T \leq \sigma^2 \log^{2\theta} \frac{eT}{\delta}$, thus we know

$$b_T \leq 2 b_0 + \frac{4(F(x_1) - F^*)}{\eta} + 4\eta L \log^+ \frac{\eta L}{b_0} + 4\sigma \sqrt{T \left( \log^{2\theta} \frac{eT}{\delta} \right) \log \left( 1 + \frac{16\sigma^2 T \log^{2\theta} \frac{eT}{\delta}}{b_0^2} \right)}.$$

$\square$

By using Lemma A.6, we can consider the following decomposition

$$\mathbb{E}\left[\Delta_T\right] = \mathbb{E}\left[\Delta_T \mathbb{1}_{E(\delta)}\right] + \mathbb{E}\left[\Delta_T \mathbb{1}_{E^c(\delta)}\right]$$

$$= \mathbb{E}\left[\frac{\Delta_T}{b_T} b_T \mathbb{1}_{E(\delta)}\right] + \mathbb{E}\left[\Delta_T \mathbb{1}_{E^c(\delta)}\right]$$

$$\leq g_T(\delta) \mathbb{E}\left[\frac{\Delta_T}{b_T} \mathbb{1}_{E(\delta)}\right] + \mathbb{E}\left[\Delta_T \mathbb{1}_{E^c(\delta)}\right]$$

$$\leq g_T(\delta) \mathbb{E}\left[\frac{\Delta_T}{b_T}\right] + \mathbb{E}\left[\Delta_T \mathbb{1}_{E^c(\delta)}\right].$$

Note that Lemma A.2 tells us

$$\mathbb{E}\left[\frac{\Delta_T}{b_T}\right] \leq \frac{\|x_1 - x^*\|^2}{\gamma\eta} + \frac{2\eta}{\gamma}\mathbb{E}\left[\frac{M_T}{b_0^2} + \log\frac{b_T}{b_0}\right].$$

Hence our remaining task is to find a proper bound on $\mathbb{E}\left[\Delta_T \mathbb{1}_{E^c(\delta)}\right]$, which is stated in the following lemma.

**Lemma A.7.** *Under Assumptions 2' and 4 we have*

$$\mathbb{E}\left[\Delta_T \mathbb{1}_{E^c(\delta)}\right] \leq \left(F(x_1) - F^* + \eta^2 L \log^+ \frac{\eta L}{2b_0}\right) T\delta + \eta\mathbb{E}^{1/4}\left[M_T^2\right]\sqrt{\log\mathbb{E}\left[\frac{b_T^2}{b_0^2}\right]}T^{3/2}\delta^{1/4}.$$

*Proof.* We restart from the smoothness of $F$:

$$F(x_{s+1}) - F(x_s) \leq -\frac{\eta}{b_s}\langle\nabla F(x_s) - \widehat{\nabla}F(x_s), \widehat{\nabla}F(x_s)\rangle - \frac{\eta}{b_s}\|\widehat{\nabla}F(x_s)\|^2 + \frac{\eta^2 L}{2b_s^2}\|\widehat{\nabla}F(x_s)\|^2.$$

Taking the sum over $s$, we have for $t \geq 2$

$$F(x_t) - F(x_1) \leq \sum_{s=1}^{t-1} -\frac{\eta}{b_s}\langle\nabla F(x_s) - \widehat{\nabla}F(x_s), \widehat{\nabla}F(x_s)\rangle + \sum_{s=1}^{t-1}\left(\frac{\eta^2 L}{2b_s^2} - \frac{\eta}{b_s}\right)\|\widehat{\nabla}F(x_s)\|^2$$

$$\leq \eta^2 L \log^+ \frac{\eta L}{2b_0} + \sum_{s=1}^{t-1}\frac{\eta}{b_s}\|\xi_s\|\|\widehat{\nabla}F(x_s)\|.$$

Following the same proof of Lemma A.6, we have

$$F(x_t) - F^* \leq F(x_1) - F^* + \eta^2 L \log^+ \frac{\eta L}{2b_0} + \eta\sqrt{M_{t-1}(t-1)\log\frac{b_{t-1}^2}{b_0^2}}.$$

Now we bound $\Delta_T$ as follows

$$\Delta_T = \sum_{t=1}^T F(x_t) - F^*$$

$$\leq F(x_1) - F^* + \sum_{t=2}^T F(x_1) - F^* + \eta^2 L \log^+ \frac{\eta L}{2b_0} + \eta\sqrt{M_{t-1}(t-1)\log\frac{b_{t-1}^2}{b_0^2}}$$

$$\leq \left(F(x_1) - F^* + \eta^2 L \log^+ \frac{\eta L}{2b_0}\right)T + \sum_{t=2}^T \eta\sqrt{M_{t-1}(t-1)\log\frac{b_{t-1}^2}{b_0^2}}$$

$$\leq \left(F(x_1) - F^* + \eta^2 L \log^+ \frac{\eta L}{2b_0}\right)T + \eta\sqrt{M_T \log\frac{b_T^2}{b_0^2}}T^{3/2}.$$

Thus we obtain

$$\mathbb{E}\left[\Delta_T \mathbb{1}_{E^c(\delta)}\right] \leq \left(F(x_1) - F^* + \eta^2 L \log^+ \frac{\eta L}{2b_0}\right)T\delta + \eta\mathbb{E}\left[\sqrt{M_T \log\frac{b_T^2}{b_0^2}}\mathbb{1}_{E^c(\delta)}\right]T^{3/2}.$$

Here we invoke Holder's inequality for three variables: for $p, q, r > 0$, $1/p + 1/q + 1/r = 1$ then $\mathbb{E}[XYZ] \leq \mathbb{E}^{1/p}[X^p]\mathbb{E}^{1/q}[Y^q]\mathbb{E}^{1/r}[Z^r]$. By substituting $X = \sqrt{M_T}$, $Y = \sqrt{\log\frac{b_T^2}{b_0^2}}$, $Z = \mathbb{1}_{E^c(\delta)}$, and $p = 4$, $q = 2$, $r = 4$, we have

$$\mathbb{E}\left[\sqrt{M_T \log\frac{b_T^2}{b_0^2}}\mathbb{1}_{E^c(\delta)}\right] \leq \mathbb{E}^{1/4}\left[M_T^2\right]\mathbb{E}^{1/2}\left[\log\frac{b_T^2}{b_0^2}\right]\mathbb{E}^{1/4}\left[\mathbb{1}_{E^c(\delta)}\right]$$

$$\leq \mathbb{E}^{1/4}\left[M_T^2\right]\sqrt{\log\mathbb{E}\left[\frac{b_T^2}{b_0^2}\right]}\delta^{1/4}.$$

So finally we get

$$\mathbb{E}\left[\Delta_T \mathbb{1}_{E^c(\delta)}\right] \leq \left(F(x_1) - F^* + \eta^2 L \log^+ \frac{\eta L}{2b_0}\right) T\delta$$

$$+ \eta \mathbb{E}^{1/4}\left[M_T^2\right] \sqrt{\log \mathbb{E}\left[\frac{b_T^2}{b_0^2}\right]} T^{3/2} \delta^{1/4}.$$

$\square$

**Lemma A.8.** *Suppose $F$ satisfies Assumptions 1, 2', 3 and 4 then*

$$\mathbb{E}\left[\Delta_T\right]$$

$$\leq g_T\left(\frac{\|x_1 - x^*\|^2}{2\gamma\eta} + \frac{2\eta\sigma^2\left(2^{(4\theta-1)\vee 2\theta}\log^{2\theta}T + C_1\right)}{\gamma b_0^2} + \frac{\eta}{\gamma}\log\mathbb{E}\left[\frac{b_T^2}{b_0^2}\right]\right)$$

$$+ \frac{F(x_1) - F^* + \eta^2 L \log^+ \frac{\eta L}{2b_0}}{T^3} + \frac{\eta\sigma\left(2^{(2\theta-1)\vee\theta}\log^{\theta}T + C_2\right)}{2}\left(1 + \log\mathbb{E}\left[\frac{b_T^2}{b_0^2}\right]\right)\sqrt{T}$$

*where $C_1 = 2^{(2\theta-1)^+}\log^{2\theta}\left(\Gamma(4\theta+1)e^2\right) + 1$ and $C_2 = 2^{(\theta-1)^+}\log^{\theta}\left(\Gamma(8\theta+1)e^2\right) + 1$ are two constants and*

$$g_T = 2b_0 + \frac{4(F(x_1) - F^*)}{\eta} + 4\eta L \log^+ \frac{\eta L}{b_0}$$

$$+ 4\sigma\sqrt{T\log^{2\theta}(eT^5)\log\left(1 + \frac{16\sigma^2 T \log^{2\theta}(eT^5)}{b_0^2}\right)}.$$

*Proof.* As stated above, we know

$$\mathbb{E}\left[\Delta_T\right] = \mathbb{E}\left[\Delta_T \mathbb{1}_{E(\delta)}\right] + \mathbb{E}\left[\Delta_T \mathbb{1}_{E^c(\delta)}\right]$$

$$= \mathbb{E}\left[\frac{\Delta_T}{b_T}b_T \mathbb{1}_{E(\delta)}\right] + \mathbb{E}\left[\Delta_T \mathbb{1}_{E^c(\delta)}\right]$$

$$\overset{(a)}{\leq} g_T(\delta)\mathbb{E}\left[\frac{\Delta_T}{b_T}\mathbb{1}_{E(\delta)}\right] + \mathbb{E}\left[\Delta_T \mathbb{1}_{E^c(\delta)}\right]$$

$$\leq g_T(\delta)\mathbb{E}\left[\frac{\Delta_T}{b_T}\right] + \mathbb{E}\left[\Delta_T \mathbb{1}_{E^c(\delta)}\right]$$

$$\overset{(b)}{\leq} g_T(\delta)\left(\frac{\|x_1 - x^*\|^2}{2\gamma\eta} + \frac{2\eta}{\gamma}\mathbb{E}\left[\frac{M_T}{b_0^2} + \log\frac{b_T}{b_0}\right]\right) + \mathbb{E}\left[\Delta_T \mathbb{1}_{E^c(\delta)}\right]$$

$$\leq g_T(\delta)\left(\frac{\|x_1 - x^*\|^2}{2\gamma\eta} + \frac{2\eta}{\gamma b_0^2}\mathbb{E}\left[M_T\right] + \frac{\eta}{\gamma}\log\mathbb{E}\left[\frac{b_T^2}{b_0^2}\right]\right) + \mathbb{E}\left[\Delta_T \mathbb{1}_{E^c(\delta)}\right]$$

where $(a)$ is due to Lemma A.6. $(b)$ is by Lemma A.2.

Lemma A.7 gives us

$$\mathbb{E}\left[\Delta_T \mathbb{1}_{E^c(\delta)}\right] \leq \left(F(x_1) - F^* + \eta^2 L \log^+ \frac{\eta L}{2b_0}\right) T\delta + \eta\mathbb{E}^{1/4}\left[M_T^2\right]\sqrt{\log\mathbb{E}\left[\frac{b_T^2}{b_0^2}\right]}T^{3/2}\delta^{1/4}.$$

Pluggin in this bound, we have

$$\mathbb{E}\left[\Delta_T\right]$$

$$\leq g_T(\delta)\left(\frac{\|x_1 - x^*\|^2}{2\gamma\eta} + \frac{2\eta}{\gamma b_0^2}\mathbb{E}\left[M_T\right] + \frac{\eta}{\gamma}\log\mathbb{E}\left[\frac{b_T^2}{b_0^2}\right]\right)$$

$$+ \left(F(x_1) - F^* + \eta^2 L \log^+ \frac{\eta L}{2b_0}\right)T\delta + \eta\mathbb{E}^{1/4}\left[M_T^2\right]\sqrt{\log\mathbb{E}\left[\frac{b_T^2}{b_0^2}\right]}T^{3/2}\delta^{1/4}.$$

Now we take $\delta = T^{-4}$ and let $g_T := g_T(T^{-4})$ to obtain

$$\mathbb{E}\left[\Delta_T\right]$$

$$\leq g_T \left( \frac{\|x_1 - x^*\|^2}{2\gamma\eta} + \frac{2\eta}{\gamma b_0^2} \mathbb{E}\left[M_T\right] + \frac{\eta}{\gamma} \log \mathbb{E}\left[\frac{b_T^2}{b_0^2}\right] \right)$$

$$+ \frac{1}{T^3} \left( F(x_1) - F^* + \eta^2 L \log^+ \frac{\eta L}{2b_0} \right) + \eta \mathbb{E}^{1/4}\left[M_T^2\right] \sqrt{\log \mathbb{E}\left[\frac{b_T^2}{b_0^2}\right]} \sqrt{T}.$$

From Lemma A.5, we know

$$\mathbb{E}\left[M_T\right] \leq \sigma^2 \left( \log^{2\theta}\left(\Gamma(4\theta+1)e^2 T^2\right) + 1 \right) \leq \sigma^2 (2^{(4\theta-1)\vee 2\theta} \log^{2\theta}(T) + C_1)$$

and

$$\mathbb{E}\left[M_T^2\right] \leq \sigma^4 (\log^{4\theta}(\Gamma(8\theta+1)e^2 T^2) + 1)$$

$$\Rightarrow \mathbb{E}^{1/4}\left[M_T^2\right] = \sigma \left( \log^{4\theta}(\Gamma(8\theta+1)e^2 T^2) + 1 \right)^{1/4} \leq \sigma (\log^\theta(\Gamma(8\theta+1)e^2 T^2) + 1)$$

$$\leq \sigma(2^{(2\theta-1)\vee\theta} \log^\theta T + C_2)$$

Hence we have

$$\mathbb{E}\left[\Delta_T\right]$$

$$\leq g_T \left( \frac{\|x_1 - x^*\|^2}{2\gamma\eta} + \frac{2\eta\sigma^2 \left( 2^{(4\theta-1)\vee 2\theta} \log^{2\theta} T + C_1 \right)}{\gamma b_0^2} + \frac{\eta}{\gamma} \log \mathbb{E}\left[\frac{b_T^2}{b_0^2}\right] \right)$$

$$+ \frac{F(x_1) - F^* + \eta^2 L \log^+ \frac{\eta L}{2b_0}}{T^3} + \eta\sigma \left( 2^{(2\theta-1)\vee\theta} \log^\theta T + C_2 \right) \sqrt{\log \mathbb{E}\left[\frac{b_T^2}{b_0^2}\right]} \sqrt{T}$$

$$\leq g_T \left( \frac{\|x_1 - x^*\|^2}{2\gamma\eta} + \frac{2\eta\sigma^2 \left( 2^{(4\theta-1)\vee 2\theta} \log^{2\theta} T + C_1 \right)}{\gamma b_0^2} + \frac{\eta}{\gamma} \log \mathbb{E}\left[\frac{b_T^2}{b_0^2}\right] \right)$$

$$+ \frac{F(x_1) - F^* + \eta^2 L \log^+ \frac{\eta L}{2b_0}}{T^3} + \frac{\eta\sigma \left( 2^{(2\theta-1)\vee\theta} \log^\theta T + C_2 \right)}{2} \left( 1 + \log \mathbb{E}\left[\frac{b_T^2}{b_0^2}\right] \right) \sqrt{T}$$

$$\square$$

With these results, we can finally show the theorem 3.5.

*Proof of Theorem 3.5 .* The key technique we use is the self-bounding argument. That is, we have expressed a bound for $\mathbb{E}[\Delta_T]$ via $\mathbb{E}[b_T^2/b_0^2]$, now we will show how to bound this term via $\Delta_T$. To do this, we rely on the smoothness assumption and Lemma A.3

$$\mathbb{E}\left[b_T^2\right] = \mathbb{E}\left[b_0^2 + \sum_{t=1}^T \|\widehat{\nabla} F(x_t)\|^2\right]$$

$$\leq b_0^2 + \mathbb{E}\left[\sum_{t=1}^T 2\|\xi_t\|^2\right] + \mathbb{E}\left[\sum_{t=1}^T 2\|\nabla F(x_t)\|^2\right]$$

$$\leq b_0^2 + 2\Gamma(2\theta+1)e\sigma^2 T + \mathbb{E}\left[4L\sum_{t=1}^T F(x_t) - F(x^*)\right]$$

$$\leq b_0^2 + 2\Gamma(2\theta+1)e\sigma^2 T + 4L\mathbb{E}\left[\Delta_T\right].$$

Thus from Lemma A.8 we can write

$$\mathbb{E}\left[\Delta_T\right] \leq G_0 + G_1 \log \left( 1 + \frac{2\Gamma(2\theta+1)e\sigma^2 T}{b_0^2} + \frac{4L}{b_0^2} \mathbb{E}\left[\Delta_T\right] \right) \tag{5}$$

where

$$G_0 = \frac{F(x_1) - F^* + \eta^2 L \log^+ \frac{\eta L}{2b_0}}{T^3} + \frac{\eta \sigma \left(2^{(2\theta-1)\vee\theta} \log^\theta T + C_2\right)\sqrt{T}}{2}$$

$$+ g_T \left(\frac{\|x_1 - x^*\|^2}{2\gamma\eta} + \frac{2\eta\sigma^2 \left(2^{(4\theta-1)\vee 2\theta} \log^{2\theta}(T) + C_1\right)}{\gamma b_0^2}\right)$$

$$= O\left(1 + \sigma\sqrt{T \log^{2\theta} T} + (1 + \sigma^2 \log^{2\theta} T)g_T\right)$$

$$G_1 = \frac{\eta\sigma\left(2^{(2\theta-1)\vee\theta} \log^\theta T + C_2\right)\sqrt{T}}{2} + \frac{\eta g_T}{\gamma}$$

$$= O\left(\sigma\sqrt{T \log^{2\theta} T} + g_T\right)$$

$$g_T = 2b_0 + \frac{4(F(x_1) - F^*)}{\eta} + 4\eta L \log^+ \frac{\eta L}{b_0} + 4\sigma\sqrt{T \log^{2\theta}(eT^5) \log\left(1 + \frac{16\sigma^2 T \log^{2\theta}(eT^5)}{b_0^2}\right)}$$

$$= O\left(1 + \sigma\sqrt{T \log^{2\theta} T \log(1 + \sigma^2 T \log^{2\theta} T)}\right)$$

Now we solve (5). Consider two cases:

If $4L\mathbb{E}[\Delta_T] \leq 2\Gamma(2\theta+1)e\sigma^2 T$ then

$$\mathbb{E}[\Delta_T] \leq G_0 + G_1 \log\left(1 + \frac{4\Gamma(2\theta+1)e\sigma^2 T}{b_0^2}\right).$$

If $4L\mathbb{E}[\Delta_T] \geq 2\Gamma(2\theta+1)e\sigma^2 T$ then

$$\mathbb{E}[\Delta_T] \leq G_0 + G_1 \log\left(1 + \frac{8L}{b_0^2}\mathbb{E}[\Delta_T]\right)$$

$$= G_0 + G_1 \log\left(\frac{1 + \frac{8L}{b_0^2}\mathbb{E}[\Delta_T]}{1 + 16LG_1/b_0^2}\right) + G_1 \log\left(1 + \frac{16LG_1}{b_0^2}\right)$$

$$\leq G_0 + G_1 \frac{1 + \frac{8L}{b_0^2}\mathbb{E}[\Delta_T]}{1 + 16LG_1/b_0^2} + G_1 \log\left(1 + \frac{16LG_1}{b_0^2}\right)$$

$$\leq G_0 + G_1 + \frac{\mathbb{E}[\Delta_T]}{2} + G_1 \log\left(1 + \frac{16LG_1}{b_0^2}\right)$$

$$\Rightarrow \mathbb{E}[\Delta_T] \leq 2G_0 + 2G_1 + 2G_1 \log\left(1 + \frac{16eLG_1}{b_0^2}\right).$$

In both cases, we have

$$\mathbb{E}[\Delta_T] \leq 3G_0 + 2G_1 + 2G_1 \log\left(1 + \frac{16eLG_1}{b_0^2}\right) + G_1 \log\left(1 + \frac{4\Gamma(2\theta+1)e\sigma^2 T}{b_0^2}\right)$$

$$= O\left((1 + \text{poly}(\sigma^2 \log^{2\theta} T, \log(1 + \sigma^2 T \log^{2\theta} T)))(1 + \sigma\sqrt{T \log^{2\theta} T})\right)$$

Dividing both sides by $T$ concludes the proof. $\qquad\square$

### A.2.3 CONVERGENCE OF STOCHASTIC ADAGRADNORM UNDER WEAKER ASSUMPTIONS

Note that Theorem 3.5 depends on the stronger Assumption 2' instead of Assumption 2. Besides, in Section 3.1, we proved that Assumptions 1 and 2 are enough to ensure that AdaGradNorm can converge in the deterministic setting. Hence it is reasonable to conjecture Stochastic AdaGradNorm can also converge if replacing Assumption 2' by Assumption 2. In this section, we show that, indeed, this conjecture is true.

**Theorem A.9.** *Suppose $F$ satisfies Assumptions 1, 2, 3 and 4. Stochastic AdaGradNorm admits*

$$\mathbb{E}\left[\sqrt{\frac{\sum_{t=1}^{T} F(x_t) - F(x^*)}{T}}\right] = O\left((1 + \mathrm{poly}(\log(1 + \sigma\sqrt{T}), \sigma^2 \log^{2\theta} T))\left(\frac{1}{\sqrt{T}} + \frac{\sigma^{1/2}}{T^{1/4}}\right)\right).$$

*Proof.* First we invoke Lemma A.2 to get

$$\mathbb{E}\left[\frac{\Delta_T}{b_T}\right] \leq \frac{\|x_1 - x^*\|^2}{\gamma\eta} + \frac{2\eta}{\gamma}\mathbb{E}\left[\frac{M_T}{b_0^2} + \log\frac{b_T}{b_0}\right].$$

Using Holder's inequality we have

$$\mathbb{E}\left[\sqrt{\Delta_T}\right] \leq \sqrt{\left(\frac{\|x_1 - x^*\|^2}{\gamma\eta} + \frac{2\eta}{\gamma}\mathbb{E}\left[\frac{M_T}{b_0^2} + \log\frac{b_T}{b_0}\right]\right)\mathbb{E}\left[b_T\right]}$$

$$\leq \sqrt{\left(\frac{\|x_1 - x^*\|^2}{\gamma\eta} + \frac{2\eta}{\gamma b_0^2}\mathbb{E}\left[M_T\right] + \frac{2\eta}{\gamma}\log\frac{\mathbb{E}\left[b_T\right]}{b_0}\right)\mathbb{E}\left[b_T\right]}.$$

Applying Lemma A.5 with $p = 1$ to get

$$\mathbb{E}\left[M_T\right] \leq \sigma^2\left(\log^{2\theta}\left(\Gamma(4\theta + 1)e^2 T^2\right) + 1\right)$$

$$\leq \sigma^2\left(2^{2\theta}\log^{2\theta}\left(T^2\right) + 2^{2\theta}\log^{2\theta}\left(\Gamma(4\theta + 1)e^2\right) + 1\right)$$

$$= \sigma^2\left(2^{4\theta}\log^{2\theta} T + C\right)$$

where $C = 2^{2\theta}\log^{2\theta}\left(\Gamma(4\theta + 1)e^2\right) + 1$.

Besides, note that

$$b_T = \sqrt{b_0^2 + \sum_{t=1}^{T}\|\widehat{\nabla}F(x_t)\|^2} \leq \sqrt{b_0^2 + 2\sum_{t=1}^{T}\|\xi_t\|^2 + 4L\Delta_T} \leq b_0 + \sqrt{2\sum_{t=1}^{T}\|\xi_t\|^2 + 2\sqrt{L\Delta_T}}.$$

Thus we know

$$\mathbb{E}\left[b_T\right] \leq \mathbb{E}\left[b_0 + \sqrt{2\sum_{t=1}^{T}\|\xi_t\|^2 + 2\sqrt{L\Delta_T}}.\right]$$

$$\leq b_0 + \sqrt{2\sum_{t=1}^{T}\mathbb{E}\left[\|\xi_t\|^2\right] + 2\sqrt{L}\mathbb{E}\left[\sqrt{\Delta_T}\right]}$$

$$\leq b_0 + \sqrt{2\Gamma(2\theta + 1)e\sigma^2 T} + 2\sqrt{L}\mathbb{E}\left[\sqrt{\Delta_T}\right]$$

where the last inequality is due to Lemma A.3.

Hence, by letting

$$B_1 = \frac{\|x_1 - x^*\|^2}{\gamma\eta} + \frac{2\eta\left(2^{4\theta}\log^{2\theta} T + C\right)\sigma^2}{\gamma b_0^2}$$

$$= O(1 + \sigma^2\log^{2\theta} T)$$

$$B_2 = b_0 + \sqrt{2\Gamma(2\theta + 1)e\sigma^2 T}$$

$$= O(1 + \sigma\sqrt{T})$$

$$X = \mathbb{E}\left[\sqrt{\Delta_T}\right]$$

we can solve the following inequality

$$X^2 \leq \left(B_1 + \frac{2\eta}{\gamma}\log\left(\frac{B_2 + 2\sqrt{L}X}{b_0}\right)\right)(B_2 + 2\sqrt{L}X)$$

to get the final result. $\qquad\square$

## A.3  ADAGRAD

---

**Algorithm 7** AdaGrad

---

Initialize: $x_1, \eta > 0$
for $t = 1$ to $T$
  for $j = 1$ to $d$
    $b_{t,j} = \sqrt{b_{0,j}^2 + \sum_{i=1}^{t} \left( \nabla_j F(x_i) \right)^2}$
    $x_{t+1,j} = x_{t,j} - \frac{\eta}{b_{t,j}} \nabla_j F(x_t)$

---

In this section, we will extend the result of AdaGradNorm to AdaGrad (Algorithm 7) in the deterministic setting. To our knowledge, we are the first to give the explicit bound of the counvergence rate of AdaGrad on $\mathbb{R}^d$. First, we examine the growth of the stepsize.

**Lemma A.10.** *Suppose $F$ satisfies Assumptions 1 and 2", we have*

$$\sum_{j=1}^{d} b_{T,j} \leq \sum_{j=1}^{d} b_{0,j} + \frac{2(F(x_1) - F^*)}{\eta} + 2\eta \sum_{j=1}^{d} L_j \log^+ \frac{\eta L_j}{b_{0,j}}.$$

*Proof.* By smoothness we have

$$F(x_{t+1}) - F(x_t) \leq \langle \nabla F(x_t), x_{t+1} - x_t \rangle + \frac{\|x_{t+1} - x_t\|_{\mathbf{L}}^2}{2}$$

$$= \sum_{j=1}^{d} \left( -\frac{\eta}{b_{t,j}} + \frac{L_j \eta^2}{2b_{t,j}^2} \right) \nabla_j F(x_t)^2$$

$$\Rightarrow \sum_{j=1}^{d} \frac{\eta}{2b_{t,j}} \nabla_j F(x_t)^2 \leq F(x_t) - F(x_{t+1}) + \sum_{j=1}^{d} \left( \frac{L_j \eta^2}{2b_{t,j}^2} - \frac{\eta}{2b_{t,j}} \right) \nabla_j F(x_t)^2$$

$$\Rightarrow \sum_{t=1}^{T} \sum_{j=1}^{d} \frac{\eta}{2b_{t,j}} \nabla_j F(x_t)^2 \leq F(x_1) - F^* + \sum_{t=1}^{T} \sum_{j=1}^{d} \left( \frac{L_j \eta^2}{2b_{t,j}^2} - \frac{\eta}{2b_{t,j}} \right) \nabla_j F(x_t)^2.$$

Note that, for the L.H.S.,

$$\sum_{t=1}^{T} \sum_{j=1}^{d} \frac{\eta}{2b_{t,j}} \nabla_j F(x_t)^2 = \frac{\eta}{2} \sum_{j=1}^{d} \sum_{t=1}^{T} \frac{b_{t,j}^2 - b_{t-1,j}^2}{b_{t,j}}$$

$$\geq \frac{\eta}{2} \sum_{j=1}^{d} \sum_{t=1}^{T} b_{t,j} - b_{t-1,j}$$

$$= \frac{\eta}{2} \sum_{j=1}^{d} \left( b_{T,j} - b_{0,j} \right).$$

Besides,

$$\sum_{t=1}^{T} \sum_{j=1}^{d} \left( \frac{L_j \eta^2}{2b_{t,j}^2} - \frac{\eta}{2b_{t,j}} \right) \nabla_j F(x_t)^2 = \frac{\eta}{2} \sum_{t=1}^{T} \sum_{j=1}^{d} \left( \frac{L_j \eta}{b_{t,j}^2} - \frac{1}{b_{t,j}} \right) \nabla_j F(x_t)^2$$

$$\leq \frac{\eta}{2} \sum_{j=1}^{d} \sum_{t=1}^{\tau_j} \frac{L_j \eta}{b_{t,j}^2} \nabla_i F(x_t)^2 \quad (\tau_j \text{ is the last } t \text{ such that} b_{t,j} \leq \eta L_j)$$

$$= \frac{\eta^2}{2} \sum_{j=1}^{d} L_j \sum_{t=1}^{\tau_i} \frac{b_{t,j}^2 - b_{t-1,j}^2}{b_{t,j}^2}$$

$$\leq \eta^2 \sum_{j=1}^{d} L_j \log^+ \frac{\eta L_j}{b_{0,j}}.$$

Hence we have

$$\sum_{j=1}^{d} b_{T,j} \le \sum_{j=1}^{d} b_{0,j} + \frac{2(F(x_1) - F^*)}{\eta} + 2\eta \sum_{j=1}^{d} L_j \log^+ \frac{\eta L_j}{b_{0,j}}.$$

$\square$

Theorem A.11 states the convergence guarantee for Algorithm 7.

**Theorem A.11.** *Suppose F satisfies Assumptions 1 and 2", we have*

$$\frac{\sum_{t=1}^{T} F(x_t) - F^*}{T}$$

$$\le \frac{\left(\sum_{j=1}^{d} b_{0,j} + \frac{2(F(x_1)-F^*)}{\eta} + 2\eta \sum_{j=1}^{d} L_j \log^+ \frac{\eta L_j}{b_{0,j}}\right)^d \left(\frac{\|x_1 - x^*\|_{\mathbf{b}_1}^2}{\gamma \eta} + \frac{2\eta}{\gamma} \sum_{j=1}^{d} \left(\frac{2\eta L_j}{\gamma} - b_{0,j}\right)^+\right)}{T \left(d^d \prod_{j=1}^{d} b_{0,j}\right)}$$

*where*

$$\|x_1 - x^*\|_{\mathbf{b}_1}^2 = \sum_{j=1}^{d} b_{1,j}(x_{1,j} - x_j^*)^2.$$

Before going into the proof, it is worth discussing the result above as well as the main challenges and differences compared with AdaGradNorm. For simplicity, let $\mathbf{b}_t = \text{diag}(b_{t,i})$. We can expect that, by a similar argument that we used before to bound the function value gap via the stepsize, we will have

$$\frac{\sum_{t=1}^{T} F(x_t) - F^*}{T} \le \frac{g(\mathbf{b}_T) \left(\frac{\|x_1 - x^*\|_{\mathbf{b}_1}^2}{\gamma \eta} + \frac{2\eta}{\gamma} \sum_{j=1}^{d} \left(\frac{2\eta L_j}{\gamma} - b_{0,j}\right)^+\right)}{T}$$

where $g(\mathbf{b}_T)$ is a function of the last stepsize and the factor $\frac{\|x_1 - x^*\|_{\mathbf{b}_1}^2}{\gamma \eta} + \frac{2\eta}{\gamma} \sum_{j=1}^{d} \left(\frac{2\eta L_j}{\gamma} - b_{0,j}\right)^+$ is obtained in a similar manner as before, but in $d$-dimensions. The challenge is that since the stepsize is a vector, it is not possible to use "division" by the stepsize as in AdaGradNorm. On the one hand, we can overcome this by rewriting the argument; on the other hand, this problem will incur an exponential rate for $g(\mathbf{b}_T)$ dependent on the smoothness parameters.

*Proof.* We can write

$$x_{t+1} = x_t - \eta \mathbf{b}_t^{-1} \nabla F(x_t).$$

Starting from $\gamma$-quasar convexity

$$F(x_t) - F^* \le \frac{\langle \nabla F(x_t), x_t - x^* \rangle}{\gamma}$$

$$= \frac{\langle \mathbf{b}_t (x_t - x_{t+1}), x_t - x^* \rangle}{\eta \gamma}$$

$$= \frac{\|x_t - x^*\|_{\mathbf{b}_t}^2 - \|x_{t+1} - x^*\|_{\mathbf{b}_t}^2 + \|x_{t+1} - x_t\|_{\mathbf{b}_t}^2}{2\eta \gamma}$$

$$= \frac{\|x_t - x^*\|_{\mathbf{b}_t}^2 - \|x_{t+1} - x^*\|_{\mathbf{b}_t}^2}{2\eta \gamma} + \frac{\eta}{2\gamma} \|\nabla F(x_t)\|_{\mathbf{b}_t^{-1}}^2.$$

Note that $F$ also satisfies Assumption 2"

$$F(x_t) - F^* \ge \frac{\|\nabla F(x_t)\|_{\mathbf{L},*}^2}{2}.$$

Hence

$$\frac{F(x_t) - F^*}{2} + \frac{\|\nabla F(x_t)\|_{\mathbf{L},*}^2}{4} \le F(x_t) - F^*$$

$$\le \frac{\|x_t - x^*\|_{\mathbf{b}_t}^2 - \|x_{t+1} - x^*\|_{\mathbf{b}_t}^2}{2\eta\gamma} + \frac{\eta}{2\gamma}\|\nabla F(x_t)\|_{\mathbf{b}_t^{-1}}^2$$

$$\Rightarrow \frac{F(x_t) - F^*}{2} \le \frac{\|x_t - x^*\|_{\mathbf{b}_t}^2 - \|x_{t+1} - x^*\|_{\mathbf{b}_t}^2}{2\eta\gamma} + \frac{\eta}{2\gamma}\|\nabla F(x_t)\|_{\mathbf{b}_t^{-1}}^2 - \frac{\|\nabla F(x_t)\|_{\mathbf{L},*}^2}{4}$$

$$\Rightarrow F(x_t) - F^* \le \frac{\|x_t - x^*\|_{\mathbf{b}_t}^2 - \|x_{t+1} - x^*\|_{\mathbf{b}_t}^2}{\eta\gamma} + \sum_{j=1}^d \left(\frac{\eta}{\gamma b_{t,j}} - \frac{1}{2L_j}\right)\nabla_j F(x_t)^2.$$

Taking the sum over $t$

$$\sum_{t=1}^T F(x_t) - F^*$$

$$\le \sum_{t=1}^T \frac{\|x_t - x^*\|_{\mathbf{b}_t}^2 - \|x_{t+1} - x^*\|_{\mathbf{b}_t}^2}{\eta\gamma} + \sum_{t=1}^T \sum_{j=1}^d \left(\frac{\eta}{\gamma b_{t,j}} - \frac{1}{2L_j}\right)\nabla_j F(x_t)^2$$

$$= \frac{\|x_1 - x^*\|_{\mathbf{b}_1}^2 - \|x_{T+1} - x^*\|_{\mathbf{b}_T}^2}{\gamma\eta} + \sum_{t=2}^T \frac{\|x_t - x^*\|_{\mathbf{b}_t - \mathbf{b}_{t-1}}^2}{\gamma\eta} + \sum_{t=1}^T \sum_{j=1}^d \left(\frac{\eta}{\gamma b_{t,j}} - \frac{1}{2L_j}\right)\nabla_j F(x_t)^2.$$

Due to the excess term $\sum_{t=2}^T \frac{\|x_t - x^*\|_{\mathbf{b}_t - \mathbf{b}_{t-1}}^2}{\gamma\eta}$ in the RHS, we need to proceed and bound $\|x_t - x^*\|_{\mathbf{b}_t - \mathbf{b}_{t-1}}^2$. First, observe that since the L.H.S. is non-negative,

$$\frac{\|x_{T+1} - x^*\|_{\mathbf{b}_T}^2}{\gamma\eta} \le \frac{\|x_1 - x^*\|_{\mathbf{b}_1}^2}{\gamma\eta} + \sum_{t=2}^T \frac{\|x_t - x^*\|_{\mathbf{b}_t - \mathbf{b}_{t-1}}^2}{\gamma\eta} + \sum_{t=1}^T \sum_{j=1}^d \left(\frac{\eta}{\gamma b_{t,j}} - \frac{1}{2L_j}\right)\nabla_j F(x_t)^2$$

To upperbound $\|x_t - x^*\|_{\mathbf{b}_t - \mathbf{b}_{t-1}}^2$, a key observation is that

$$\|x_{T+1} - x^*\|_{\mathbf{b}_T}^2 = \|x_{T+1} - x^*\|_{\mathbf{b}_{T+1} - \mathbf{b}_T}^2 \times \frac{\|x_{T+1} - x^*\|_{\mathbf{b}_T}^2}{\|x_{T+1} - x^*\|_{\mathbf{b}_{T+1} - \mathbf{b}_T}^2}$$

$$\ge \|x_{T+1} - x^*\|_{\mathbf{b}_{T+1} - \mathbf{b}_T}^2 \min_k \frac{b_{T,k}}{b_{T+1,k} - b_{T,k}}$$

Hence for $T \ge 1$

$$\frac{\|x_{T+1} - x^*\|_{\mathbf{b}_{T+1} - \mathbf{b}_T}^2}{\gamma\eta}$$

$$\le \max_k \left(\frac{b_{T+1,k}}{b_{T,k}} - 1\right) \left[\frac{\|x_1 - x^*\|_{\mathbf{b}_1}^2}{\gamma\eta} + \sum_{t=2}^T \frac{\|x_t - x^*\|_{\mathbf{b}_t - \mathbf{b}_{t-1}}^2}{\gamma\eta} + \sum_{t=1}^T \sum_{j=1}^d \left(\frac{\eta}{\gamma b_{t,j}} - \frac{1}{2L_j}\right)\nabla_j F(x_t)^2\right]$$

By using this bound for the last term $\frac{\|x_T - x^*\|^2_{\mathbf{b}_T - \mathbf{b}_{T-1}}}{\gamma\eta}$ we obtain

$$\sum_{t=1}^{T} F(x_t) - F^*$$

$$\leq \frac{\|x_1 - x^*\|^2_{\mathbf{b}_1} - \|x_{T+1} - x^*\|^2_{\mathbf{b}_T}}{\gamma\eta} + \sum_{t=2}^{T} \frac{\|x_t - x^*\|^2_{\mathbf{b}_t - \mathbf{b}_{t-1}}}{\gamma\eta} + \sum_{t=1}^{T}\sum_{j=1}^{d}\left(\frac{\eta}{\gamma b_{t,j}} - \frac{1}{2L_j}\right)\nabla_j F(x_t)^2$$

$$\leq \frac{\|x_1 - x^*\|^2_{\mathbf{b}_1}}{\gamma\eta} + \sum_{t=2}^{T-1} \frac{\|x_t - x^*\|^2_{\mathbf{b}_t - \mathbf{b}_{t-1}}}{\gamma\eta} + \sum_{t=1}^{T}\sum_{j=1}^{d}\left(\frac{\eta}{\gamma b_{t,j}} - \frac{1}{2L_j}\right)\nabla_j F(x_t)^2$$

$$+ \max_k\left(\frac{b_{T,k}}{b_{T-1,k}} - 1\right)\left(\frac{\|x_1 - x^*\|^2_{\mathbf{b}_1}}{\gamma\eta} + \sum_{t=2}^{T-1} \frac{\|x_t - x^*\|^2_{\mathbf{b}_t - \mathbf{b}_{t-1}}}{\gamma\eta} + \sum_{t=1}^{T-1}\sum_{j=1}^{d}\left(\frac{\eta}{\gamma b_{t,j}} - \frac{1}{2L_j}\right)\nabla_j F(x_t)^2\right)$$

$$= \max_k\frac{b_{T,k}}{b_{T-1,k}}\left(\frac{\|x_1 - x^*\|^2_{\mathbf{b}_1}}{\gamma\eta} + \sum_{t=2}^{T-1} \frac{\|x_t - x^*\|^2_{\mathbf{b}_t - \mathbf{b}_{t-1}}}{\gamma\eta} + \sum_{t=1}^{T-1}\sum_{j=1}^{d}\left(\frac{\eta}{\gamma b_{t,j}} - \frac{1}{2L_j}\right)\nabla_j F(x_t)^2\right)$$

$$+ \sum_{j=1}^{d}\left(\frac{\eta}{\gamma b_{T,j}} - \frac{1}{2L_j}\right)\nabla_j F(x_T)^2$$

Continue to unroll this relation and for convenience let $\prod_{t=T+1}^{T}\max_k\frac{b_{t,k}}{b_{t-1,k}} = 1$, we have

$$\sum_{t=1}^{T} F(x_t) - F^*$$

$$\leq \left[\prod_{t=2}^{T}\max_k\frac{b_{t,k}}{b_{t-1,k}}\right]\frac{\|x_1 - x^*\|^2_{\mathbf{b}_1}}{\gamma\eta} + \sum_{t=1}^{T}\left(\prod_{\ell=t+1}^{T}\max_k\frac{b_{\ell,k}}{b_{\ell-1,k}}\right)\left(\sum_{j=1}^{d}\left(\frac{\eta}{\gamma b_{t,j}} - \frac{1}{2L_j}\right)\nabla_j F(x_t)^2\right)$$

$$= \left[\prod_{t=2}^{T}\max_k\frac{b_{t,k}}{b_{t-1,k}}\right]\frac{\|x_1 - x^*\|^2_{\mathbf{b}_1}}{\gamma\eta} + \sum_{j=1}^{d}\sum_{t=1}^{T}\left(\prod_{\ell=t+1}^{T}\max_k\frac{b_{\ell,k}}{b_{\ell-1,k}}\right)\left(\frac{\eta}{\gamma b_{t,j}} - \frac{1}{2L_j}\right)\nabla_j F(x_t)^2$$

Given $j$, if $b_{1,j} > \frac{2\eta L_j}{\gamma}$, we know

$$\sum_{t=1}^{T}\left(\prod_{\ell=t+1}^{T}\max_k\frac{b_{\ell,k}}{b_{\ell-1,k}}\right)\left(\frac{\eta}{\gamma b_{t,j}} - \frac{1}{2L_j}\right)\nabla_j F(x_t)^2 < 0 \leq \frac{2\eta}{\gamma}\left[\prod_{t=2}^{T}\max_k\frac{b_{t,k}}{b_{t-1,k}}\right]\left(\frac{2\eta L_j}{\gamma} - b_{0,j}\right)^+.$$

Otherwise, let $\tau_j$ be the last $t$ such that $b_{t,j} \leq \frac{2\eta L_j}{\gamma}$, we also have

$$
\sum_{t=1}^{T} \left( \prod_{\ell=t+1}^{T} \max_k \frac{b_{\ell,k}}{b_{\ell-1,k}} \right) \left( \frac{\eta}{\gamma b_{t,j}} - \frac{1}{2L_j} \right) \nabla_j F(x_t)^2
$$

$$
\leq \sum_{t=1}^{\tau_j} \left( \prod_{\ell=t+1}^{T} \max_k \frac{b_{\ell,k}}{b_{\ell-1,k}} \right) \left( \frac{\eta}{\gamma b_{t,j}} - \frac{1}{2L_j} \right) \nabla_j F(x_t)^2
$$

$$
\leq \left[ \prod_{t=2}^{T} \max_k \frac{b_{t,k}}{b_{t-1,k}} \right] \left( \sum_{t=1}^{\tau_j} \left( \frac{\eta}{\gamma b_{t,j}} - \frac{1}{2L_j} \right) \nabla_j F(x_t)^2 \right)
$$

$$
\leq \left[ \prod_{t=2}^{T} \max_k \frac{b_{t,k}}{b_{t-1,k}} \right] \left( \sum_{t=1}^{\tau_j} \frac{\eta}{\gamma b_{t,j}} \nabla_j F(x_t)^2 \right)
$$

$$
= \left[ \prod_{t=2}^{T} \max_k \frac{b_{t,k}}{b_{t-1,k}} \right] \left( \frac{\eta}{\gamma} \sum_{t=1}^{\tau_j} \frac{b_{t,j}^2 - b_{t-1,j}^2}{b_{t,j}} \right)
$$

$$
\leq \left[ \prod_{t=2}^{T} \max_k \frac{b_{t,k}}{b_{t-1,k}} \right] \left( \frac{2\eta}{\gamma} \sum_{t=1}^{\tau_j} b_{t,j} - b_{t-1,j} \right)
$$

$$
\leq \frac{2\eta}{\gamma} \left[ \prod_{t=2}^{T} \max_k \frac{b_{t,k}}{b_{t-1,k}} \right] \left( \frac{2\eta L_j}{\gamma} - b_{0,j} \right)^+ .
$$

Hence we know

$$
\sum_{j=1}^{d} \sum_{t=1}^{T} \left( \prod_{\ell=t+1}^{T} \max_k \frac{b_{\ell,k}}{b_{\ell-1,k}} \right) \left( \frac{\eta}{\gamma b_{t,j}} - \frac{1}{2L_j} \right) \nabla_j F(x_t)^2
$$

$$
\leq \frac{2\eta}{\gamma} \left[ \prod_{t=2}^{T} \max_k \frac{b_{t,k}}{b_{t-1,k}} \right] \left[ \sum_{j=1}^{d} \left( \frac{2\eta L_j}{\gamma} - b_{0,j} \right)^+ \right].
$$

Thus we have

$$
\sum_{t=1}^{T} F(x_t) - F^*
$$

$$
\leq \left[ \prod_{t=2}^{T} \max_k \frac{b_{t,k}}{b_{t-1,k}} \right] \frac{\|x_1 - x^*\|_{\mathbf{b}_1}^2}{\gamma \eta} + \sum_{j=1}^{d} \sum_{t=1}^{T} \left( \prod_{\ell=t+1}^{T} \max_k \frac{b_{\ell,k}}{b_{\ell-1,k}} \right) \left( \frac{\eta}{\gamma b_{t,j}} - \frac{1}{2L_j} \right) \nabla_j F(x_t)^2
$$

$$
\leq \left[ \prod_{t=2}^{T} \max_k \frac{b_{t,k}}{b_{t-1,k}} \right] \left( \frac{\|x_1 - x^*\|_{\mathbf{b}_1}^2}{\gamma \eta} + \frac{2\eta}{\gamma} \sum_{j=1}^{d} \left( \frac{2\eta L_j}{\gamma} - b_{0,j} \right)^+ \right)
$$

From Lemma A.10

$$
\sum_{j=1}^{d} b_{T,j} \leq \sum_{j=1}^{d} b_{0,j} + \frac{2(F(x_1) - F^*)}{\eta} + 2\eta \sum_{j=1}^{d} L_j \log^+ \frac{\eta L_j}{b_{0,j}}
$$

Using AM-GM we have

$$
\prod_{j=1}^{d} b_{T,j} \leq \left( \frac{\sum_{j=1}^{d} b_{T,j}}{d} \right)^d \leq \frac{1}{d^d} \left( \sum_{j=1}^{d} b_{0,j} + \frac{2(F(x_1) - F^*)}{\eta} + 2\eta \sum_{j=1}^{d} L_j \log^+ \frac{\eta L_j}{b_{0,j}} \right)^d
$$

Note that

$$
\prod_{t=2}^{T} \max_{j} \frac{b_{t,j}}{b_{t-1,j}} \leq \prod_{t=2}^{T} \prod_{j=1}^{d} \frac{b_{t,j}}{b_{t-1,j}}
$$

$$
\leq \prod_{j=1}^{d} \frac{b_{T,j}}{b_{0,j}}
$$

$$
\leq \frac{1}{d^d \prod_{j=1}^{d} b_{0,j}} \left( \sum_{j=1}^{d} b_{0,j} + \frac{2(F(x_1) - F^*)}{\eta} + 2\eta \sum_{j=1}^{d} L_j \log^+ \frac{\eta L_j}{b_{0,j}} \right)^d
$$

Hence

$$
\sum_{t=1}^{T} F(x_t) - F^*
$$

$$
\leq \left[ \prod_{t=2}^{T} \max_{j} \frac{b_{T,j}}{b_{T-1,j}} \right] \left( \frac{\|x_1 - x^*\|_{\mathbf{b}_1}^2}{\gamma \eta} + \frac{2\eta}{\gamma} \sum_{j=1}^{d} \left( \frac{2\eta L_j}{\gamma} - b_{0,j} \right)^+ \right)
$$

$$
\leq \frac{\left( \sum_{j=1}^{d} b_{0,j} + \frac{2(F(x_1)-F^*)}{\eta} + 2\eta \sum_{j=1}^{d} L_j \log^+ \frac{\eta L_j}{b_{0,j}} \right)^d \left( \frac{\|x_1-x^*\|_{\mathbf{b}_1}^2}{\gamma \eta} + \frac{2\eta}{\gamma} \sum_{j=1}^{d} \left( \frac{2\eta L_j}{\gamma} - b_{0,j} \right)^+ \right)}{d^d \prod_{j=1}^{d} b_{0,j}}
$$

which finishes the proof. $\qquad \square$

# B  MISSING PROOFS FROM SECTION 4

## B.1  IMPORTANT LEMMA

First, we state a general lemma that can be used for a more general setting. The proof of the lemma is standard.

**Lemma B.1.** *Suppose $F$ satisfies Assumptions 1 and 2' and the following conditions hold:*

- *$x_t$ is generated by $x_{t+1} = x_t - \frac{\eta}{c_t} \nabla F(x_t)$, with $\eta > 0$ and $c_t > 0$ is non-decreasing;*

- *$p_t \in (0,1]$ satisfies $\frac{1}{p_t} \geq \frac{1-p_{t+1}}{p_{t+1}}, p_1 = 1$;*

*Then we have*

$$
\frac{F(x_{T+1}) - F^*}{p_T c_T} \leq \frac{\|x_1 - x^*\|^2}{\gamma \eta} + \sum_{t=1}^{T} \left( \frac{L}{2c_t} - \frac{1}{\eta} + \frac{p_t}{\gamma \eta} - \frac{p_t c_t}{2\eta^2 L} \right) \frac{\eta^2 \|\nabla F(x_t)\|^2}{c_t^2 p_t}. \tag{6}
$$

*Proof.* Starting from $L$-smoothness

$$F(x_{t+1}) - F(x_t) \leq \langle \nabla F(x_t), x_{t+1} - x_t \rangle + \frac{L}{2} \|x_{t+1} - x_t\|^2$$

$$= \frac{2p_t}{\gamma} \langle \nabla F(x_t), x_{t+1} - x_t \rangle + \left(1 - \frac{2p_t}{\gamma}\right) \langle \nabla F(x_t), x_{t+1} - x_t \rangle + \frac{L}{2} \|x_{t+1} - x_t\|^2$$

$$= \frac{2p_t}{\gamma} \langle \nabla F(x_t), x^* - x_t \rangle + \frac{2p_t}{\gamma} \langle \nabla F(x_t), x_{t+1} - x^* \rangle$$

$$+ \left(1 - \frac{2p_t}{\gamma}\right) \langle \nabla F(x_t), x_{t+1} - x_t \rangle + \frac{L}{2} \|x_{t+1} - x_t\|^2$$

$$\leq 2p_t (F^* - F(x_t)) + \frac{p_t c_t}{\gamma \eta} \left[ \|x_t - x^*\|^2 - \|x_{t+1} - x^*\|^2 - \|x_{t+1} - x\|^2 \right]$$

$$- \left(1 - \frac{2p_t}{\gamma}\right) \frac{c_t}{\eta} \|x_{t+1} - x_t\|^2 + \frac{L}{2} \|x_{t+1} - x_t\|^2$$

$$= 2p_t (F^* - F(x_t)) + \frac{p_t c_t}{\gamma \eta} \left[ \|x_t - x^*\|^2 - \|x_{t+1} - x^*\|^2 \right]$$

$$+ \left( \frac{L}{2} - \frac{c_t}{\eta} + \frac{p_t c_t}{\gamma \eta} \right) \|x_{t+1} - x_t\|^2.$$

Note that Assumption 2 can be implied by Assumption 2', hence we have

$$F^* - F(x_t) \leq -\frac{\|\nabla F(x_t)\|^2}{2L} = -\frac{c_t^2 \|x_{t+1} - x_t\|^2}{2\eta^2 L}.$$

Therefore

$$F(x_{t+1}) - F(x_t) \leq 2p_t (F^* - F(x_t)) + \frac{p_t c_t}{\gamma \eta} \left[ \|x_t - x^*\|^2 - \|x_{t+1} - x^*\|^2 \right]$$

$$+ \left( \frac{L}{2} - \frac{c_t}{\eta} + \frac{p_t c_t}{\gamma \eta} \right) \|x_{t+1} - x_t\|^2$$

$$\leq p_t (F^* - F(x_t)) - \frac{p_t c_t^2 \|x_{t+1} - x_t\|^2}{2\eta^2 L} + \frac{p_t c_t}{\gamma \eta} \left[ \|x_t - x^*\|^2 - \|x_{t+1} - x^*\|^2 \right]$$

$$+ \left( \frac{L}{2} - \frac{c_t}{\eta} + \frac{p_t c_t}{\gamma \eta} \right) \|x_{t+1} - x_t\|^2$$

$$= p_t (F^* - F(x_t)) + \frac{p_t c_t}{\gamma \eta} \left[ \|x_t - x^*\|^2 - \|x_{t+1} - x^*\|^2 \right]$$

$$+ \left( \frac{L}{2} - \frac{c_t}{\eta} + \frac{p_t c_t}{\gamma \eta} - \frac{p_t c_t^2}{2\eta^2 L} \right) \|x_{t+1} - x_t\|^2.$$

We obtain

$$\frac{F(x_{t+1}) - F^*}{p_t c_t} \leq \frac{1 - p_t}{p_t c_t} (F(x_t) - F^*) + \frac{\|x_t - x^*\|^2 - \|x_{t+1} - x^*\|^2}{\gamma \eta}$$

$$+ \left( \frac{L}{2c_t} - \frac{1}{\eta} + \frac{p_t}{\gamma \eta} - \frac{p_t c_t}{2\eta^2 L} \right) \frac{\|x_{t+1} - x_t\|^2}{p_t}.$$

Note that we require $\frac{1}{p_t} \geq \frac{1 - p_{t+1}}{p_{t+1}}$ and $c_t$ is increasing hence

$$\frac{1}{p_t c_t} \geq \frac{1 - p_{t+1}}{p_{t+1} c_t} \geq \frac{1 - p_{t+1}}{p_{t+1} c_{t+1}}$$

which leads to

$$\frac{F(x_{T+1}) - F^*}{p_T c_T}$$

$$\leq \frac{1 - p_1}{p_1 c_1} (F(x_1) - F^*) + \frac{\|x_1 - x^*\|^2}{\gamma \eta} + \sum_{t=1}^{T} \left( \frac{L}{2c_t} - \frac{1}{\eta} + \frac{p_t}{\gamma \eta} - \frac{p_t c_t}{2\eta^2 L} \right) \frac{\|x_{t+1} - x_t\|^2}{p_t}$$

$$= \frac{1 - p_1}{p_1 c_1} (F(x_1) - F^*) + \frac{\|x_1 - x^*\|^2}{\gamma \eta} + \sum_{t=1}^{T} \left( \frac{L}{2c_t} - \frac{1}{\eta} + \frac{p_t}{\gamma \eta} - \frac{p_t c_t}{2\eta^2 L} \right) \frac{\eta^2 \|\nabla F(x_t)\|^2}{c_t^2 p_t}.$$

By setting $p_1 = 1$ we get the desired result. $\qquad\square$

## B.2 FIRST VARIANT

Note that if we assume $p_t$ satisfies the condition in Lemma B.1, by replacing $c_t$ by $b_t$, we have

$$\frac{F(x_{T+1}) - F^*}{p_T b_T} \le \frac{\|x_1 - x^*\|^2}{\gamma\eta} + \sum_{t=1}^{T}\left(\frac{L}{2b_t} - \frac{1}{\eta} + \frac{p_t}{\gamma\eta} - \frac{p_t b_t}{2\eta^2 L}\right)\frac{\eta^2\|\nabla F(x_t)\|^2}{b_t^2 p_t}$$

immediately. Now our two left tasks are to bound the residual term $\sum_{t=1}^{T}\left(\frac{L}{2b_t} - \frac{1}{\eta} + \frac{p_t}{\gamma\eta} - \frac{p_t b_t}{2\eta^2 L}\right)\frac{\eta^2\|\nabla F(x_t)\|^2}{b_t^2 p_t}$ and find an upper bound on $b_T$. Lemmas B.2 and B.3 demonstrate how we achieve these two goals.

**Lemma B.2.** *If $p_t \le 1$ for every t, we have*

$$\sum_{t=1}^{T}\left(\frac{L}{2b_t} - \frac{1}{2\eta}\right)\frac{\eta^2\|\nabla F(x_t)\|^2}{b_t^2 p_t} \le h(\Delta)$$

$$\sum_{t=1}^{T}\left(\frac{p_t}{\gamma\eta} - \frac{p_t b_t}{2\eta^2 L}\right)\frac{\eta^2\|\nabla F(x_t)\|^2}{b_t^2 p_t} \le g(\Delta)$$

*where*

$$h(\Delta) := \begin{cases} \frac{(2+\Delta)\eta(\eta L)^\Delta}{2}\log^+\frac{\eta L}{b_0} & \Delta \ge 1 \\ \frac{(2+\Delta)\eta^2 L}{2b_0^{1-\Delta}}\log^+\frac{\eta L}{b_0} & \Delta \in (0,1) \end{cases} \quad and \quad g(\Delta) := \frac{(2+\Delta)\eta}{\gamma}\left(\frac{2\eta L}{\gamma}\right)^\Delta\log^+\frac{2\eta L}{\gamma b_0}.$$

*Proof.* We first bound

$$\sum_{t=1}^{T}\left(\frac{L}{2b_t} - \frac{1}{2\eta}\right)\frac{\eta^2\|\nabla F(x_t)\|^2}{b_t^2 p_t}.$$

If $b_1 > \eta L$, we know

$$\sum_{t=1}^{T}\left(\frac{L}{2b_t} - \frac{1}{2\eta}\right)\frac{\eta^2\|\nabla F(x_t)\|^2}{b_t^2 p_t} < 0 \le h(\Delta).$$

Otherwise, we define the time $\tau = \max\{t \in [T], b_t \le \eta L\}$. Hence, we have

$$\sum_{t=1}^{T}\left(\frac{L}{2b_t} - \frac{1}{2\eta}\right)\frac{\eta^2\|\nabla F(x_t)\|^2}{b_t^2 p_t} = \sum_{t=1}^{\tau}\left(\frac{L}{2b_t} - \frac{1}{2\eta}\right)\frac{\eta^2\|\nabla F(x_t)\|^2}{b_t^2 p_t} + \sum_{t=\tau}^{T}\left(\frac{L}{2b_t} - \frac{1}{2\eta}\right)\frac{\eta^2\|\nabla F(x_t)\|^2}{b_t^2 p_t}$$

$$\le \sum_{t=1}^{\tau}\left(\frac{L}{2b_t} - \frac{1}{2\eta}\right)\frac{\eta^2\|\nabla F(x_t)\|^2}{b_t^2 p_t} \le \sum_{t=1}^{\tau}\frac{L}{2b_t} \times \frac{\eta^2\|\nabla F(x_t)\|^2}{b_t^2 p_t}$$

$$= \frac{\eta^2 L}{2}\sum_{t=1}^{\tau}\frac{b_t^{2+\Delta} - b_{t-1}^{2+\Delta}}{b_t^3} = \frac{\eta^2 L}{2}\sum_{t=1}^{\tau}\frac{b_t^{2+\Delta} - b_{t-1}^{2+\Delta}}{b_t^{2+\Delta}} \times b_t^{\Delta-1}$$

$$\le \begin{cases} \frac{\eta^2 L}{2}\sum_{t=1}^{\tau}\frac{b_t^{2+\Delta} - b_{t-1}^{2+\Delta}}{b_t^{2+\Delta}} \times (\eta L)^{\Delta-1} & \Delta \ge 1 \\ \frac{\eta^2 L}{2}\sum_{t=1}^{\tau}\frac{b_t^{2+\Delta} - b_{t-1}^{2+\Delta}}{b_t^{2+\Delta}} \times \frac{1}{b_0^{1-\Delta}} & \Delta < 1 \end{cases}$$

$$= \begin{cases} \frac{\eta(\eta L)^\Delta}{2}\sum_{t=1}^{\tau}\frac{b_t^{2+\Delta} - b_{t-1}^{2+\Delta}}{b_t^{2+\Delta}} & \Delta \ge 1 \\ \frac{\eta^2 L}{2b_0^{1-\Delta}}\sum_{t=1}^{\tau}\frac{b_t^{2+\Delta} - b_{t-1}^{2+\Delta}}{b_t^{2+\Delta}} & \Delta < 1 \end{cases}$$

$$\le \begin{cases} \frac{(2+\Delta)\eta(\eta L)^\Delta}{2}\log\frac{\eta L}{b_0} & \Delta \ge 1 \\ \frac{(2+\Delta)\eta^2 L}{2b_0^{1-\Delta}}\log\frac{\eta L}{b_0} & \Delta < 1 \end{cases}$$

$$\le h(\Delta).$$

By applying a similar argument, we can prove

$$\sum_{t=1}^{T} \left( \frac{p_t}{\gamma\eta} - \frac{p_t b_t}{2\eta^2 L} \right) \frac{\eta^2 \|\nabla F(x_t)\|^2}{b_t^2 p_t} \leq g(\Delta).$$

$\square$

**Lemma B.3.** *Suppose all the conditions in Lemma B.1 are satisfied by replacing $c_t$ by $b_t$, additionally, assume $p_t \leq 1$, we will have*

$$b_T \leq \left( \frac{2}{\eta} \left( \frac{\|x_1 - x^*\|^2}{\gamma\eta} + h(\Delta) + g(\Delta) \right) + b_0^\Delta \right)^{\frac{1}{\Delta}}$$

*Proof.* Using Lemma B.1 by replacing $c_t$ by $b_t$, we know

$$\frac{F(x_{T+1}) - F^*}{p_T b_T}$$

$$\leq \frac{\|x_1 - x^*\|^2}{\gamma\eta} + \sum_{t=1}^{T} \left( \frac{L}{2b_t} - \frac{1}{\eta} + \frac{p_t}{\gamma\eta} - \frac{p_t b_t}{2\eta^2 L} \right) \frac{\eta^2 \|\nabla F(x_t)\|^2}{b_t^2 p_t}$$

$$= \frac{\|x_1 - x^*\|^2}{\gamma\eta} + \sum_{t=1}^{T} \left( \frac{L}{2b_t} - \frac{1}{2\eta} + \frac{p_t}{\gamma\eta} - \frac{p_t b_t}{2\eta^2 L} \right) \frac{\eta^2 \|\nabla F(x_t)\|^2}{b_t^2 p_t} - \frac{\eta \|\nabla F(x_t)\|^2}{2b_t^2 p_t}$$

$$\leq \frac{\|x_1 - x^*\|^2}{\gamma\eta} + h(\Delta) + g(\Delta) - \sum_{t=1}^{T} \frac{\eta \|\nabla F(x_t)\|^2}{2b_t^2 p_t},$$

where the last inequality is by Lemma B.2. Noticing $F(x_{T+1}) - F^* \geq 0$, we know

$$\sum_{t=1}^{T} \frac{\eta \|\nabla F(x_t)\|^2}{2b_t^2 p_t} \leq \frac{\|x_1 - x^*\|^2}{\gamma\eta} + h(\Delta) + g(\Delta).$$

Now we use the update rule of $b_t$ to get

$$\sum_{t=1}^{T} \frac{\eta \|\nabla F(x_t)\|^2}{2b_t^2 p_t} = \frac{\eta}{2} \sum_{t=1}^{T} \frac{b_t^{2+\Delta} - b_{t-1}^{2+\Delta}}{b_t^2} \geq \frac{\eta}{2} \sum_{t=1}^{T} b_t^\Delta - b_{t-1}^\Delta = \frac{\eta}{2} \left( b_T^\Delta - b_0^\Delta \right)$$

Hence we know

$$b_T \leq \left( \frac{2}{\eta} \left( \frac{\|x_1 - x^*\|^2}{\gamma\eta} + h(\Delta) + g(\Delta) \right) + b_0^\Delta \right)^{\frac{1}{\Delta}}$$

$\square$

Equipped with Lemmas B.2 and B.3, we can give a proof of Theorem 4.1.

*Proof.* Note that if $p_t = \frac{1}{t}$, all the conditions in Lemma B.1 are satisfied by replacing $c_t$ by $b_t$. Hence we have

$$\frac{F(x_{T+1}) - F^*}{p_T b_T} \leq \frac{\|x_1 - x^*\|^2}{\gamma\eta} + \sum_{t=1}^{T} \left( \frac{L}{2b_t} - \frac{1}{\eta} + \frac{p_t}{\gamma\eta} - \frac{p_t b_t}{2\eta^2 L} \right) \frac{\eta^2 \|\nabla F(x_t)\|^2}{b_t^2 p_t}$$

$$\leq \frac{\|x_1 - x^*\|^2}{\gamma\eta} + \sum_{t=1}^{T} \left( \frac{L}{2b_t} - \frac{1}{2\eta} + \frac{p_t}{\gamma\eta} - \frac{p_t b_t}{2\eta^2 L} \right) \frac{\eta^2 \|\nabla F(x_t)\|^2}{b_t^2 p_t}$$

$$\leq \frac{\|x_1 - x^*\|^2}{\gamma\eta} + h(\Delta) + g(\Delta),$$

where the last inequality is by Lemma B.2. Multiplying both sides by $p_T b_T$, we get

$$F(x_{T+1}) - F^* \leq \frac{b_T \left( \frac{\|x_1 - x^*\|^2}{\gamma\eta} + h(\Delta) + g(\Delta) \right)}{T}.$$

By using the upper bound of $b_T$ in Lemma B.3, we finish the proof. $\square$

### B.3 SECOND VARIANT

Similar to the previous section, what we need to do is to bound the residual term $\sum_{t=1}^{T} \left( \frac{L}{2c_t} - \frac{1}{\eta} + \frac{p_t}{\gamma\eta} - \frac{p_t c_t}{2\eta^2 L} \right) \frac{\eta^2 \|\nabla F(x_t)\|^2}{c_t^2 p_t}$ and find an upper bound on $c_T$ where $c_t = b_t^{\delta} b_{t-1}^{1-\delta}$ here. We first bound the residual term by the following lemma.

**Lemma B.4.** *If $p_t \leq 1$ for every $t$, we have*

$$\sum_{t=1}^{T} \left( \frac{L}{2c_t} - \frac{1}{2\eta} \right) \frac{\eta^2 \|\nabla F(x_t)\|^2}{c_t^2 p_t} \leq \frac{\eta^2 L}{b_0} \left( 1 - \left( \frac{b_0}{\eta L} \right)^{\frac{1}{\delta}} \right)^+$$

$$\sum_{t=1}^{T} \left( \frac{p_t}{\gamma\eta} - \frac{p_t c_t}{2\eta^2 L} \right) \frac{\eta^2 \|\nabla F(x_t)\|^2}{c_t^2 p_t} \leq \frac{2\eta}{\gamma\delta} \left( \frac{2\eta L}{\gamma b_0} \right)^{\frac{2}{\delta}-2} \log^+ \frac{2\eta L}{\gamma b_0}$$

*where $c_t = b_t^{\delta} b_{t-1}^{1-\delta}$.*

*Proof.* Note that $\frac{c_t}{c_{t-1}} = \frac{b_t^{\delta}}{b_{t-1}^{2\delta-1} b_{t-2}^{1-\delta}} \geq 1$, this means $c_t$ is monotone increasing. We first bound

$$\sum_{t=1}^{T} \left( \frac{L}{2c_t} - \frac{1}{2\eta} \right) \frac{\eta^2 \|\nabla F(x_t)\|^2}{c_t^2 p_t}.$$

If $c_1 > \eta L$, we know

$$\sum_{t=1}^{T} \left( \frac{L}{2c_t} - \frac{1}{2\eta} \right) \frac{\eta^2 \|\nabla F(x_t)\|^2}{c_t^2 p_t} < 0 \leq \frac{\eta^2 L}{b_0} \left( 1 - \left( \frac{b_0}{\eta L} \right)^{\frac{1}{\delta}} \right)^+.$$

Otherwise, let $\tau = \max\{t \in [T], c_t \leq \eta L\}$. We have

$$\sum_{t=1}^{T} \left( \frac{L}{2c_t} - \frac{1}{2\eta} \right) \frac{\eta^2 \|\nabla F(x_t)\|^2}{c_t^2 p_t} \leq \sum_{t=1}^{\tau} \frac{\eta^2 L \|\nabla F(x_t)\|^2}{2 c_t^3 p_t}$$

$$= \frac{\eta^2 L}{2} \sum_{t=1}^{\tau} \frac{b_t^2 - b_{t-1}^2}{b_t^{3\delta} b_{t-1}^{3-3\delta}}$$

$$\leq \eta^2 L \sum_{t=1}^{\tau} \frac{b_t - b_{t-1}}{b_t^{3\delta-1} b_{t-1}^{3-3\delta}}$$

$$\leq \eta^2 L \sum_{t=1}^{\tau} \frac{b_t - b_{t-1}}{b_t b_{t-1}}$$

$$\leq \eta^2 L \left( \frac{1}{b_0} - \frac{1}{b_\tau} \right).$$

Note that $c_\tau = b_\tau^{\delta} b_{\tau-1}^{1-\delta} \leq \eta L \Rightarrow b_\tau \leq \left( \frac{\eta L}{b_{\tau-1}^{1-\delta}} \right)^{1/\delta}$. Hence

$$\frac{1}{b_0} - \frac{1}{b_\tau} \leq \frac{1}{b_0} - \frac{(b_{\tau-1})^{\frac{1}{\delta}-1}}{(\eta L)^{1/\delta}} \leq \frac{1}{b_0} \left( 1 - \left( \frac{b_0}{\eta L} \right)^{\frac{1}{\delta}} \right)^+.$$

Combining two cases, there is always

$$\sum_{t=1}^{T} \left( \frac{L}{2c_t} - \frac{1}{2\eta} \right) \frac{\eta^2 \|\nabla F(x_t)\|^2}{c_t^2 p_t} \leq \frac{\eta^2 L}{b_0} \left( 1 - \left( \frac{b_0}{\eta L} \right)^{\frac{1}{\delta}} \right)^+.$$

Now we turn to the second bound. If $c_1 > 2\eta L/\gamma$, we know

$$\sum_{t=1}^{T} \left( \frac{p_t}{\gamma\eta} - \frac{p_t c_t}{2\eta^2 L} \right) \frac{\eta^2 \|\nabla F(x_t)\|^2}{c_t^2 p_t} < 0 \leq \frac{2\eta}{\gamma\delta} \left( \frac{2\eta L}{\gamma b_0} \right)^{\frac{2}{\delta}-2} \log^+ \frac{2\eta L}{\gamma b_0}.$$

Otherwise, we define the time $\tau = \max\{t \in [T], c_t \leq 2\eta L/\gamma\}$. Then, we have

$$\sum_{t=1}^{T}\left(\frac{p_t}{\gamma\eta} - \frac{p_t c_t}{2\eta^2 L}\right)\frac{\eta^2\|\nabla F(x_t)\|^2}{c_t^2 p_t} \leq \sum_{t=1}^{\tau}\frac{p_t}{\gamma\eta}\frac{\eta^2\|\nabla F(x_t)\|^2}{c_t^2 p_t}$$

$$\leq \frac{\eta}{\gamma}\sum_{t=1}^{\tau}\frac{\|\nabla F(x_t)\|^2}{c_t^2 p_t}$$

$$= \frac{\eta}{\gamma}\sum_{t=1}^{\tau}\frac{b_t^2 - b_{t-1}^2}{b_t^{2\delta}b_{t-1}^{2-2\delta}}$$

$$= \frac{\eta}{\gamma}\sum_{t=1}^{\tau}\left(\frac{b_t}{b_{t-1}}\right)^{2-2\delta}\frac{b_t^2 - b_{t-1}^2}{b_t^2}$$

Because $b_t^\delta b_{t-1}^{1-\delta} = c_t \leq 2\eta L/\gamma$ for $t \leq \tau$, so we know $b_t \leq \left(\frac{2\eta L}{\gamma b_{t-1}^{1-\delta}}\right)^{1/\delta}$. Using this bound

$$\frac{\eta}{\gamma}\sum_{t=1}^{\tau}\left(\frac{b_t}{b_{t-1}}\right)^{2-2\delta}\frac{b_t^2 - b_{t-1}^2}{b_t^2} \leq \frac{\eta}{\gamma}\sum_{t=1}^{\tau}\left(\frac{2\eta L}{\gamma b_{t-1}}\right)^{\frac{2}{\delta}-2}\frac{b_t^2 - b_{t-1}^2}{b_t^2}$$

$$\leq \frac{\eta}{\gamma}\left(\frac{2\eta L}{\gamma b_0}\right)^{\frac{2}{\delta}-2}\sum_{t=1}^{\tau}\frac{b_t^2 - b_{t-1}^2}{b_t^2}$$

$$\leq \frac{2\eta}{\gamma}\left(\frac{2\eta L}{\gamma b_0}\right)^{\frac{2}{\delta}-2}\log\frac{b_t}{b_0}$$

$$\leq \frac{2\eta}{\gamma\delta}\left(\frac{2\eta L}{\gamma b_0}\right)^{\frac{2}{\delta}-2}\log^+\frac{2\eta L}{\gamma b_0}.$$

The proof is completed. $\qquad\square$

As before, our last task is to bound $c_T$. It is enough to bound $b_T$ since $c_T \leq b_T$.

**Lemma B.5.** *Suppose all the conditions in Lemma B.1 are satisfied by replacing $c_t$ by $b_t$, additionally, assume $p_t \leq 1$, we will have*

$$b_T \leq b_0 \exp\left(\frac{\frac{\|x_1 - x^*\|^2}{\gamma\eta^2} + \frac{\eta L}{b_0}\left(1 - \left(\frac{b_0}{\eta L}\right)^{\frac{1}{\delta}}\right)^+ + \frac{2}{\gamma\delta}\left(\frac{2\eta L}{\gamma b_0}\right)^{\frac{2}{\delta}-2}\log^+\frac{2\eta L}{\gamma b_0}}{1-\delta}\right).$$

*Proof.* By Lemma B.1, we know

$$0 \leq \frac{F(x_{T+1}) - F^*}{p_T c_T}$$

$$\leq \frac{\|x_1 - x^*\|^2}{\gamma\eta} + \sum_{t=1}^{T}\left(\frac{L}{2c_t} - \frac{1}{\eta} + \frac{p_t}{\gamma\eta} - \frac{p_t c_t}{2\eta^2 L}\right)\frac{\eta^2\|\nabla F(x_t)\|^2}{c_t^2 p_t}$$

$$0 \leq \frac{\|x_1 - x^*\|^2}{\gamma\eta} + \sum_{t=1}^{T}\left(\frac{L}{2c_t} - \frac{1}{2\eta} + \frac{p_t}{\gamma\eta} - \frac{p_t c_t}{2\eta^2 L}\right)\frac{\eta^2\|\nabla F(x_t)\|^2}{c_t^2 p_t} - \frac{\eta\|\nabla F(x_t)\|^2}{2c_t^2 p_t}$$

$$\sum_{t=1}^{T}\frac{\eta\|\nabla F(x_t)\|^2}{2c_t^2 p_t} \leq \frac{\|x_1 - x^*\|^2}{\gamma\eta} + \sum_{t=1}^{T}\left(\frac{L}{2c_t} - \frac{1}{2\eta} + \frac{p_t}{\gamma\eta} - \frac{p_t c_t}{2\eta^2 L}\right)\frac{\eta^2\|\nabla F(x_t)\|^2}{c_t^2 p_t}$$

$$\leq \frac{\|x_1 - x^*\|^2}{\gamma\eta} + \frac{\eta^2 L}{b_0}\left(1 - \left(\frac{b_0}{\eta L}\right)^{\frac{1}{\delta}}\right)^+ + \frac{2\eta}{\gamma\delta}\left(\frac{2\eta L}{\gamma b_0}\right)^{\frac{2}{\delta}-2}\log^+\frac{2\eta L}{\gamma b_0}$$

where the last inequality is by Lemma B.4. Note that for the L.H.S., we have

$$\sum_{t=1}^{T} \frac{\eta \|\nabla F(x_t)\|^2}{2c_t^2 p_t} = \frac{\eta}{2} \sum_{t=1}^{T} \frac{b_t^2 - b_{t-1}^2}{b_t^{2\delta} b_{t-1}^{2-2\delta}} = \frac{\eta}{2} \sum_{t=1}^{T} \left( \frac{b_t}{b_{t-1}} \right)^{2-2\delta} - \left( \frac{b_{t-1}}{b_t} \right)^{2\delta}$$

$$\geq \eta(1-\delta) \sum_{t=1}^{T} \log \frac{b_t}{b_{t-1}} = \eta(1-\delta) \log \frac{b_T}{b_0}.$$

Hence we know

$$b_T \leq b_0 \exp \left( \frac{\frac{\|x_1 - x^*\|^2}{\gamma \eta^2} + \frac{\eta L}{b_0} \left( 1 - \left( \frac{b_0}{\eta L} \right)^{\frac{1}{\delta}} \right)^{+} + \frac{2}{\gamma \delta} \left( \frac{2\eta L}{\gamma b_0} \right)^{\frac{2}{\delta} - 2} \log^{+} \frac{2\eta L}{\gamma b_0}}{1 - \delta} \right)$$

$\square$

Finally, the proof of Theorem 4.3 is similar to the proof of Theorem 4.1, hence, which is omitted.

## B.4 An asymptotic rate when $\Delta = 0$ and $\delta = 1$

As mentioned before, by setting $\Delta = 0$ in Algorithm 3 and $\delta = 1$ in Algorithm 4 we obtain the same algorithm. The square root update rule of $b_t$ and the step size now are both more similar to the original AdaGradNorm. Intuitively, we can also expect the convergence of the last iterate in this case; furthermore, by taking the limit when $\Delta \to 0$ and $\delta \to 1$, we can have a sense of the exponential dependency of the provable convergence rate on the problem parameters. However, previous analysis strictly requires that $\Delta > 0$ and $\delta < 1$, thus does not apply here.

In this section, we partially confirm the convergence of this variant by proving an asymptotic rate, i.e., $F(x_{T+1}) - F^* = O(1/T)$. Unfortunately, under Assumptions 1 and 2', we cannot figure out the explicit dependency of the convergence rate on the problem parameters. However, in the next section, we will give an explicit rate by replacing Assumption 1 with the stronger Assumption 1'. As stated, our goal is to prove Theorem B.6 in this section.

**Theorem B.6.** *Suppose $F$ satisfies Assumptions 1 and 2', when $\Delta = 0$ for Algorithm 3, or equivalently, $\delta = 1$ for Algorithm 4, by taking $p_t = \frac{1}{t}$, we have*

$$F(x_{T+1}) - F^* = O(1/T).$$

Before starting the proof, we first discuss why we can obtain only an asymptotic rate when $\Delta = 0$ and $\delta = 1$. As before, one can still expect that $F(x_{T+1}) - F^* \leq \frac{b_T C}{T}$ remains true for some constant $C$. However, a critical difference will show up when we want to find an explicit upper bound on $b_T$. Using the proof of Lemma B.3 as an example (similarly for the proof of Lemma B.5), one key step is to get $\sum_{t=1}^{T} \frac{\|\nabla F(x_t)\|^2}{b_t^2 p_t} = O(1)$, where in the previous analysis, by replacing $\frac{\|\nabla F(x_t)\|^2}{p_t}$ by $b_t^{2+\Delta} - b_{t-1}^{2+\Delta}$ with $\Delta > 0$, we can lower bound $\sum_{t=1}^{T} \frac{\|\nabla F(x_t)\|^2}{b_t^2 p_t}$ by a function of $b_T$ and finally give an explicit bound on $b_T$. However, this is not possible when $\Delta = 0$ as $\sum_{t=1}^{T} \frac{\|\nabla F(x_t)\|^2}{b_t^2 p_t} = \sum_{t=1}^{T} \frac{b_t^2 - b_{t-1}^2}{b_t^2}$. The only information we can get from $\sum_{t=1}^{T} \frac{b_t^2 - b_{t-1}^2}{b_t^2} = O(1)$ is $\lim_{T\to\infty} \frac{b_{T-1}^2}{b_T^2} = 1$. This is not enough to tell us whether $b_T$ is upper bounded or not. In Lemma B.8, we will use a new argument to finally show that $\lim_{T\to\infty} b_T < \infty$, which leads to an asymptotic rate as desired. It is worth pointing out that finding an asymptotic without explicit dependency on the problem parameters is the approach used in some of the previous work, such as Antonakopoulos et al. (2022). This also gives us a glimpse of the method used to analyze the convergence of the accelerated methods in Section 5.

Now we start the proof. As before, we can employ Lemma B.1. Hence we only need to bound the residual terms as following

**Lemma B.7.** *Suppose $p_t \leq 1$, when $\Delta = 0$ for Algorithm 3, or equivalently, $\delta = 1$ for Algorithm 4, we have*

$$\sum_{t=1}^{T} \left( \frac{L}{2b_t} - \frac{1}{2\eta} \right) \frac{\eta^2 \|\nabla F(x_t)\|^2}{b_t^2 p_t} \leq \eta \left( \frac{\eta L}{b_0} - 1 \right)^+$$

$$\sum_{t=1}^{T} \left( \frac{p_t}{\gamma \eta} - \frac{p_t b_t}{2\eta^2 L} \right) \frac{\eta^2 \|\nabla F(x_t)\|^2}{b_t^2 p_t} \leq \frac{2\eta}{\gamma} \log^+ \frac{2\eta L}{\gamma b_0}$$

The proof is essentially similar to the proof of Lemmas B.2 and B.4, hence we omit it here.

**Lemma B.8.** *Suppose all the conditions in Lemma B.1 are satisfied by replacing $c_t$ by $b_t$, then when $\Delta = 0$ for Algorithm 3, or equivalently, $\delta = 1$ for Algorithm 4, we have*

$$\lim_{T \to \infty} b_T = b_\infty < \infty.$$

*Proof.* First note that $b_t$ is increasing, by the Monotone convergence theorem, we know $\lim_{T \to \infty} b_T = b_\infty$ exists. We aim to show $b_\infty < \infty$. By Lemma B.1 and replacing $c_t$ by $b_t$, we have

$$\frac{F(x_{T+1}) - F^*}{p_T b_T}$$

$$\leq \frac{\|x_1 - x^*\|^2}{\gamma \eta} + \sum_{t=1}^{T} \left( \frac{L}{2b_t} - \frac{1}{\eta} + \frac{p_t}{\gamma \eta} - \frac{p_t b_t}{2\eta^2 L} \right) \frac{\eta^2 \|\nabla F(x_t)\|^2}{b_t^2 p_t}$$

$$= \frac{\|x_1 - x^*\|^2}{\gamma \eta} + \sum_{t=1}^{T} \left( \frac{L}{2b_t} - \frac{1}{2\eta} + \frac{p_t}{\gamma \eta} - \frac{p_t b_t}{2\eta^2 L} \right) \frac{\eta^2 \|\nabla F(x_t)\|^2}{b_t^2 p_t} - \frac{1}{2\eta} \times \frac{\eta^2 \|\nabla F(x_t)\|^2}{b_t^2 p_t}$$

$$= \frac{\|x_1 - x^*\|^2}{\gamma \eta} + \sum_{t=1}^{T} \left( \frac{L}{2b_t} - \frac{1}{2\eta} + \frac{p_t}{\gamma \eta} - \frac{p_t b_t}{2\eta^2 L} \right) \frac{\eta^2 \|\nabla F(x_t)\|^2}{b_t^2 p_t} - \frac{\eta \|\nabla F(x_t)\|^2}{2b_t^2 p_t}$$

$$\leq \frac{\|x_1 - x^*\|^2}{\gamma \eta} + \eta \left( \frac{\eta L}{b_0} - 1 \right)^+ + \frac{2\eta}{\gamma} \log^+ \frac{2\eta L}{\gamma b_0} - \sum_{t=1}^{T} \frac{\eta \|\nabla F(x_t)\|^2}{2b_t^2 p_t},$$

where the last inequality is by Lemma B.7. Noticing $F(x_{T+1}) - F^* \geq 0$, we know

$$\sum_{t=1}^{T} \frac{\eta \|\nabla F(x_t)\|^2}{2b_t^2 p_t} \leq \frac{\|x_1 - x^*\|^2}{\gamma \eta} + \eta \left( \frac{\eta L}{b_0} - 1 \right)^+ + \frac{2\eta}{\gamma} \log^+ \frac{2\eta L}{\gamma b_0},$$

which implies

$$\sum_{t=1}^{\infty} \frac{\|\nabla F(x_t)\|^2}{b_t^2 p_t} \leq \frac{2\|x_1 - x^*\|^2}{\gamma \eta^2} + 2 \left( \frac{\eta L}{b_0} - 1 \right)^+ + \frac{4}{\gamma} \log^+ \frac{2\eta L}{\gamma b_0}. \tag{7}$$

We observe that

$$b_T^2 = b_{T-1}^2 + \frac{\|\nabla F(x_T)\|^2}{p_T}$$

$$\Rightarrow b_T^2 = \frac{b_{T-1}^2}{1 - \frac{\|\nabla F(x_T)\|^2}{b_T^2 p_T}} = b_0^2 \prod_{t=1}^{T} \frac{1}{1 - \frac{\|\nabla F(x_t)\|^2}{b_t^2 p_t}}.$$

Taking log to both sides, we get

$$\log b_T^2 = \log b_0^2 + \sum_{t=1}^{T} \log \frac{1}{1 - \frac{\|\nabla F(x_t)\|^2}{b_t^2 p_t}} \leq \log b_0^2 + \sum_{t=1}^{T} \frac{1}{1 - \frac{\|\nabla F(x_t)\|^2}{b_t^2 p_t}} - 1$$

$$= \log b_0^2 + \sum_{t=1}^{T} \frac{\frac{\|\nabla F(x_t)\|^2}{b_t^2 p_t}}{1 - \frac{\|\nabla F(x_t)\|^2}{b_t^2 p_t}} \leq \log b_0^2 + \sum_{t=1}^{\infty} \frac{\frac{\|\nabla F(x_t)\|^2}{b_t^2 p_t}}{1 - \frac{\|\nabla F(x_t)\|^2}{b_t^2 p_t}}$$

Note that Inequality (7) tells us $\lim_{t\to\infty} \frac{\|\nabla F(x_t)\|^2}{b_t^2 p_t} = 0$, hence we can let $\tau$ be the time such that $\frac{\|\nabla F(x_t)\|^2}{b_t^2 p_t} \leq \frac{1}{2}$ for $t \geq \tau$. Then we know

$$\sum_{t=1}^{\infty} \frac{\frac{\|\nabla F(x_t)\|^2}{b_t^2 p_t}}{1 - \frac{\|\nabla F(x_t)\|^2}{b_t^2 p_t}} = \sum_{t=1}^{\tau-1} \frac{\frac{\|\nabla F(x_t)\|^2}{b_t^2 p_t}}{1 - \frac{\|\nabla F(x_t)\|^2}{b_t^2 p_t}} + \sum_{t=\tau}^{\infty} \frac{\frac{\|\nabla F(x_t)\|^2}{b_t^2 p_t}}{1 - \frac{\|\nabla F(x_t)\|^2}{b_t^2 p_t}}$$

$$\leq \sum_{t=1}^{\tau-1} \frac{\frac{\|\nabla F(x_t)\|^2}{b_t^2 p_t}}{1 - \frac{\|\nabla F(x_t)\|^2}{b_t^2 p_t}} + 2\sum_{t=\tau}^{\infty} \frac{\|\nabla F(x_t)\|^2}{b_t^2 p_t}$$

$$\leq \sum_{t=1}^{\tau-1} \frac{\frac{\|\nabla F(x_t)\|^2}{b_t^2 p_t}}{1 - \frac{\|\nabla F(x_t)\|^2}{b_t^2 p_t}} + 2\left(\frac{2\|x_1 - x^*\|^2}{\gamma\eta^2} + 2\left(\frac{\eta L}{b_0} - 1\right)^+ + \frac{4}{\gamma}\log^+\frac{2\eta L}{\gamma b_0}\right)$$

$$< \infty.$$

The above result implies $\log b_T^2$ has a uniform upper bound which means $b_\infty < \infty$. $\qquad\square$

Now we can start to prove Theorem B.6.

*Proof.* Note that when $\Delta = 0$ for Algorithm 3, or equivalently, $\delta = 1$ for Algorithm 4, if $p_t = \frac{1}{t}$, all the conditions in Lemma B.1 are satisfied by replacing $c_t$ by $b_t$. Hence we have

$$\frac{F(x_{T+1}) - F^*}{p_T b_T} \leq \frac{\|x_1 - x^*\|^2}{\gamma\eta} + \sum_{t=1}^{T}\left(\frac{L}{2b_t} - \frac{1}{\eta} + \frac{p_t}{\gamma\eta} - \frac{p_t b_t}{2\eta^2 L}\right)\frac{\eta^2\|\nabla F(x_t)\|^2}{b_t^2 p_t}$$

$$\leq \frac{\|x_1 - x^*\|^2}{\gamma\eta} + \sum_{t=1}^{T}\left(\frac{L}{2b_t} - \frac{1}{2\eta} + \frac{p_t}{\gamma\eta} - \frac{p_t b_t}{2\eta^2 L}\right)\frac{\eta^2\|\nabla F(x_t)\|^2}{b_t^2 p_t}$$

$$= \frac{\|x_1 - x^*\|^2}{\gamma\eta} + \eta\left(\frac{\eta L}{b_0} - 1\right)^+ + \frac{2\eta}{\gamma}\log^+\frac{2\eta L}{\gamma b_0},$$

where the last inequality is by Lemma B.7. Multiplying both sides by $p_T b_T$, we wknow

$$F(x_{T+1}) - F^* \leq \frac{b_T}{T}\left(\frac{\|x_1 - x^*\|^2}{\gamma\eta} + \eta\left(\frac{\eta L}{b_0} - 1\right)^+ + \frac{2\eta}{\gamma}\log^+\frac{2\eta L}{\gamma b_0}\right)$$

$$\leq \frac{b_\infty}{T}\left(\frac{\|x_1 - x^*\|^2}{\gamma\eta} + \eta\left(\frac{\eta L}{b_0} - 1\right)^+ + \frac{2\eta}{\gamma}\log^+\frac{2\eta L}{\gamma b_0}\right)$$

$$= O\left(\frac{1}{T}\right),$$

where the last line is by Lemma B.8. $\qquad\square$

## B.5 A NON-ASYMPTOTIC RATE WHEN $\Delta = 0$ AND $\delta = 1$ FOR CONVEX SMOOTH FUNCTIONS

In the previous section, we only give an asymptotic rate when $\Delta = 0$ and $\delta = 1$. In the following, we will show that, by replacing Assumption 1 by the stronger Assumption 1', a non-asymptotic rate can be obtained as stated in Theorem B.9.

**Theorem B.9.** *Suppose $F$ satisfies Assumptions 1' and 2', when $\Delta = 0$ for Algorithm 3, or equivalently, $\delta = 1$ for Algorithm 4, by taking $p_t = \frac{1}{t}$, we have*

$$F(x_{T+1}) - F^* \leq \frac{b\left(\frac{\|x_1 - x^*\|^2}{2\eta} + \frac{\eta}{2}\left(\frac{2\eta L}{b_0} - 1\right)^+\right)}{T},$$

*where* $b = \max\left\{\frac{\eta L}{2}, \sqrt{b_0^2 + \|\nabla F(x_1)\|^2}\exp\left(\frac{3\|x_1 - x^*\|^2}{\eta^2} + 3\left(\frac{2\eta L}{b_0} - 1\right)^+\right),\right.$

$\left.\eta L\sqrt{\frac{1}{4} + \frac{\|x_t - x^*\|^2}{\eta^2} + \left(\frac{2\eta L}{b_0} - 1\right)^+}\exp\left(\frac{3\|x_1 - x^*\|^2}{\eta^2} + 3\left(\frac{2\eta L}{b_0} - 1\right)^+\right)\right\}.$

We first give another well-known characterization of convex and $L$-smooth functions without proof.

**Lemma B.10.** *Suppose $F$ satisfies Assumption 1' and 2', then $\forall x, y \in \mathbb{R}^d$*

$$\langle \nabla F(x) - \nabla F(y), x - y \rangle \geq \frac{\|\nabla F(x) - \nabla F(y)\|^2}{L}.$$

Next, we state a simple variant of Lemma B.1, the proof of which is essentially the same as the proof of Lemma B.1, hence we omit it.

**Lemma B.11.** *Suppose the following conditions hold:*

- *$F$ satisfies Assumptions 1' and 2';*

- *$p_t \in (0, 1)$ satisfies $\frac{1}{p_t} \geq \frac{1 - p_{t+1}}{p_{t+1}}, p_1 = 1.$*

*When $\Delta = 0$ for Algorithm 3, or equivalently, $\delta = 1$ for Algorithm 4, we have*

$$\frac{F(x_{T+1}) - F^*}{p_T b_T} \leq \frac{\|x_1 - x^*\|^2}{2\eta} + \sum_{t=1}^{T} \left( \frac{L}{2b_t} - \frac{1}{\eta} + \frac{p_t}{2\eta} \right) \frac{\eta^2 \|\nabla F(x_t)\|^2}{b_t^2 p_t}$$

The same as Lemma B.7, we give the following bound on the residual term without proof.

**Lemma B.12.** *Suppose $p_t \leq 1$, when $\Delta = 0$ for Algorithm 3, or equivalently, $\delta = 1$ for Algorithm 4, we have*

$$\sum_{t=1}^{T} \left( \frac{L}{2b_t} - \frac{1}{4\eta} \right) \frac{\eta^2 \|\nabla F(x_t)\|^2}{b_t^2 p_t} \leq \frac{\eta}{2} \left( \frac{2\eta L}{b_0} - 1 \right)^+$$

Again, the above two lemmas give us

$$F(x_{T+1}) - F^* \leq p_T b_T \left( \frac{\|x_1 - x^*\|^2}{2\eta} + \frac{\eta}{2} \left( \frac{2\eta L}{b_0} - 1 \right)^+ \right) \tag{8}$$

W.l.o.g., we assume $b_T > \frac{\eta L}{2}$ in the following analysis. Otherwise, we can use the bound $b_T \leq \frac{\eta L}{2}$ to get a trivial convergence rate. Now we define the time

$$\tau = \max \left\{ t \in [T], b_t \leq \frac{\eta L}{2} \right\} \vee 0.$$

This time $\tau$ is extremly useful and will finally help us bound $b_T$. Now we list the following three important lemmas related to time $\tau$.

**Lemma B.13.** *With Assumptions 1' and 2', when $t \geq \tau + 1$, $\|\nabla F(x_t)\|$ is non-increasing.*

*Proof.* Taking $x = x_t, y = x_{t+1}$ in Lemma B.10, we get

$$\frac{\|\nabla F(x_t) - \nabla F(x_{t+1})\|^2}{L} \leq \langle \nabla F(x_t) - \nabla F(x_{t+1}), x_t - x_{t+1} \rangle$$

$$= \langle \nabla F(x_t) - \nabla F(x_{t+1}), \frac{\eta}{b_t} \nabla F(x_t) \rangle$$

$$\Rightarrow \left( \frac{1}{L} - \frac{\eta}{b_t} \right) \|\nabla F(x_t)\|^2 + \frac{1}{L} \|\nabla F(x_{t+1})\|^2 \leq \left( \frac{2}{L} - \frac{\eta}{b_t} \right) \langle \nabla F(x_t), \nabla F(x_{t+1}) \rangle.$$

Note that when $t \geq \tau + 1$, we know $b_t > \frac{\eta L}{2} \Rightarrow \frac{2}{L} - \frac{\eta}{b_t} > 0$, hence we have

$$\left( \frac{1}{L} - \frac{\eta}{b_t} \right) \|\nabla F(x_t)\|^2 + \frac{1}{L} \|\nabla F(x_{t+1})\|^2$$

$$\leq \left( \frac{2}{L} - \frac{\eta}{b_t} \right) \langle \nabla F(x_t), \nabla F(x_{t+1}) \rangle$$

$$\leq \left( \frac{1}{L} - \frac{\eta}{2b_t} \right) \|\nabla F(x_t)\|^2 + \left( \frac{1}{L} - \frac{\eta}{2b_t} \right) \|\nabla F(x_{t+1})\|^2,$$

which implies $\|\nabla F(x_{t+1})\|^2 \leq \|\nabla F(x_t)\|^2$. This is just what we want. $\qquad \square$

**Lemma B.14.** *With Assumptions 1' and 2', if $p_t = \frac{1}{t}$, when $t \geq \tau + 2 \geq 2$,*

$$\frac{\|\nabla F(x_t)\|^2}{b_t^2 p_t} \leq \frac{2}{3}$$

*Proof.* This is because

$$\begin{aligned}
\frac{\|\nabla F(x_t)\|^2}{b_t^2 p_t} &= \frac{t\|\nabla F(x_t)\|^2}{b_0^2 + \sum_{i=1}^t i\|\nabla F(x_i)\|^2} \\
&\leq \frac{t\|\nabla F(x_t)\|^2}{(t-1)\|\nabla F(x_{t-1})\|^2 + t\|\nabla F(x_t)\|^2} \\
&\leq \frac{t\|\nabla F(x_t)\|^2}{(t-1)\|\nabla F(x_t)\|^2 + t\|\nabla F(x_t)\|^2} \\
&= \frac{t}{2t-1},
\end{aligned}$$

where the last inequality is because $t - 1 \geq \tau + 1$, hence $\|\nabla F(x_{t-1})\| \geq \|\nabla F(x_t)\|$ by Lemma B.13. Note that $t \geq 2$, so $\frac{\|\nabla F(x_t)\|^2}{b_t^2 p_t} \leq \frac{t}{2t-1} \leq \frac{2}{3}$. $\qquad\square$

**Lemma B.15.** *With Assumptions 1' and 2', if $p_t = \frac{1}{t}$*

$$b_{\tau+1} \leq \sqrt{b_0^2 + \|\nabla F(x_1)\|^2} \vee \eta L \sqrt{\frac{1}{4} + \frac{\|x_1 - x^*\|^2}{\eta^2} + \left(\frac{2\eta L}{b_0} - 1\right)^+}$$

*Proof.* If $\tau = 0$, we have

$$b_{\tau+1} = b_1 = \sqrt{b_0^2 + \|\nabla F(x_1)\|^2}.$$

Otherwise, we know $\tau + 1 \geq 2$, hence

$$\begin{aligned}
b_{\tau+1}^2 &= b_\tau^2 + (\tau+1)\|\nabla F(x_{\tau+1})\|^2 \\
&\leq b_\tau^2 + 2L(\tau+1)(F(x_{\tau+1}) - F^*) \\
&\leq b_\tau^2 + 2L\frac{\tau+1}{\tau}b_\tau\left(\frac{\|x_1 - x^*\|^2}{2\eta} + \frac{\eta}{2}\left(\frac{2\eta L}{b_0} - 1\right)^+\right) \\
&\leq b_\tau^2 + 4Lb_\tau\left(\frac{\|x_1 - x^*\|^2}{2\eta} + \frac{\eta}{2}\left(\frac{2\eta L}{b_0} - 1\right)^+\right) \\
&\leq \left(\frac{\eta L}{2}\right)^2 + \eta^2 L^2\left(\frac{\|x_1 - x^*\|^2}{\eta^2} + \left(\frac{2\eta L}{b_0} - 1\right)^+\right) \\
\Rightarrow b_{\tau+1} &\leq \eta L\sqrt{\frac{1}{4} + \frac{\|x_1 - x^*\|^2}{\eta^2} + \left(\frac{2\eta L}{b_0} - 1\right)^+}
\end{aligned}$$

where the second inequality is due to (8). $\qquad\square$

Now we combine Lemmas B.14 and B.15 to get an upper bound for $b_T$.

**Lemma B.16.** *With Assumptions 1' and 2', if $p_t = \frac{1}{t}$*

$$\begin{aligned}
b_T \leq \max\Bigg\{ &\frac{\eta L}{2}, \sqrt{b_0^2 + \|\nabla F(x_1)\|^2}\exp\left(\frac{3\|x_1 - x^*\|^2}{\eta^2} + 3\left(\frac{2\eta L}{b_0} - 1\right)^+\right), \\
&\eta L\sqrt{\frac{1}{4} + \frac{\|x_1 - x^*\|^2}{\eta^2} + \left(\frac{2\eta L}{b_0} - 1\right)^+}\exp\left(\frac{3\|x_1 - x^*\|^2}{\eta^2} + 3\left(\frac{2\eta L}{b_0} - 1\right)^+\right)\Bigg\}
\end{aligned}$$

*Proof.* Note that if $b_T \leq \frac{\eta L}{2}$, we are done. If $b_T > \frac{\eta L}{2}$, we will bound $b_T$ as follows:

$$b_T^2 = b_{T-1}^2 + \frac{\|\nabla F(x_T)\|^2}{p_T}$$

$$\Rightarrow b_T^2 = \frac{b_{T-1}^2}{1 - \frac{\|\nabla F(x_T)\|^2}{b_T^2 p_T}} = b_{\tau+1}^2 \prod_{t=\tau+2}^{T} \frac{1}{1 - \frac{\|\nabla F(x_t)\|^2}{b_t^2 p_t}}$$

$$\Rightarrow \log b_T^2 \leq \log b_{\tau+1}^2 + \sum_{t=\tau+2}^{T} \log \frac{1}{1 - \frac{\|\nabla F(x)\|^2}{b_t^2 p_t}}$$

$$\leq \log b_{\tau+1}^2 + \sum_{t=\tau+2}^{T} \frac{\frac{\|\nabla F(x)\|^2}{b_t^2 p_t}}{1 - \frac{\|\nabla F(x)\|^2}{b_t^2 p_t}}$$

$$\leq \log b_{\tau+1}^2 + \sum_{t=\tau+2}^{T} \frac{3\|\nabla F(x)\|^2}{b_t^2 p_t},$$

where the last inequality is by Lemma B.14. Noticing $p_t = \frac{1}{t} \leq 1$, combining Lemmas B.11 and B.12, we can find

$$\sum_{t=1}^{T} \frac{\|\nabla F(x_t)\|^2}{b_t^2 p_t} \leq \frac{2\|x_1 - x^*\|^2}{\eta^2} + 2 \left( \frac{2\eta L}{b_0} - 1 \right)^+ .$$

Hence we know

$$\log b_T^2 \leq \log b_{\tau+1}^2 + \sum_{t=\tau+2}^{T} \frac{3\|\nabla F(x)\|^2}{b_t^2 p_t}$$

$$\leq \log b_{\tau+1}^2 + \frac{6\|x_1 - x^*\|^2}{\eta^2} + 6 \left( \frac{2\eta L}{b_0} - 1 \right)^+$$

$$\Rightarrow b_T^2 \leq b_{\tau+1}^2 \exp \left( \frac{6\|x_1 - x^*\|^2}{\eta^2} + 6 \left( \frac{2\eta L}{b_0} - 1 \right)^+ \right).$$

The last step is to use the bound on $b_{\tau+1}$ in Lemma B.15. $\qquad \square$

Finally, the proof of Theorem B.9 is obtained by simply using Lemma B.16 to Equation (8).

## C  MISSING PROOFS FROM SECTION 5

### C.1  IMPORTANT LEMMA

First, we state a general lemma that can be used for a more general setting. The proof of the lemma is standard.

**Lemma C.1.** *Suppose $F$ satisfies Assumptions 1' and 2' and the following conditions hold:*

* $w_t$ *is generated by*

$$v_t = (1 - a_t)w_t + a_t x_t$$

$$x_{t+1} = x_t - \frac{\eta}{q_t c_t} \nabla F(v_t)$$

$$w_{t+1} = (1 - a_t)w_t + a_t x_{t+1}$$

  *with $\eta > 0$ and $c_t > 0$ is non-decreasing;*

* $a_t \in (0, 1]$ *and $q_t \geq a_t$ satisfy $\frac{1}{a_t q_t} \geq \frac{1 - a_{t+1}}{a_{t+1} q_{t+1}}, a_1 = 1$;*

*Then we have*

$$\frac{F(w_{T+1}) - F^*}{a_T q_T c_T} \leq \frac{\|x_1 - x^*\|^2}{2\eta} + \sum_{t=1}^{T} \left( \frac{L}{2b_t} - \frac{1}{2\eta} \right) \frac{\eta^2 \|\nabla F(v_t)\|^2}{c_t^2 q_t^2}$$

*Proof.* Starting from smoothness

$$F(w_{t+1}) - F(v_t)$$

$$\leq \langle \nabla F(v_t), w_{t+1} - v_t \rangle + \frac{L}{2} \|w_{t+1} - v_t\|^2$$

$$= (1 - a_t)\langle \nabla F(v_t), w_t - v_t \rangle + a_t \langle \nabla F(v_t), x_{t+1} - v_t \rangle + \frac{L}{2} \|w_{t+1} - v_t\|^2$$

$$= (1 - a_t)\langle \nabla F(v_t), w_t - v_t \rangle + a_t \langle \nabla F(v_t), x^* - v_t \rangle + a_t \langle \nabla F(v_t), x_{t+1} - x^* \rangle + \frac{L}{2} \|w_{t+1} - v_t\|^2$$

$$\leq (1 - a_t)(F(w_t) - F(v_t)) + a_t(F^* - F(v_t)) + a_t \langle \nabla F(v_t), x_{t+1} - x^* \rangle + \frac{L}{2} \|w_{t+1} - v_t\|^2$$

Thus

$$F(w_{t+1}) - F^* \leq (1 - a_t)(F(w_t) - F^*) + a_t \langle \nabla F(v_t), x_{t+1} - x^* \rangle + \frac{L}{2} \|w_{t+1} - v_t\|^2$$

where the last inequality is due to the convexity of $F$. Using the update rule $\nabla F(v_t) = \frac{q_t c_t}{\eta}(x_t - x_{t+1})$ and $w_{t+1} - v_t = a_t(x_{t+1} - x_t)$ we obtain

$$F(w_{t+1}) - F^* \leq (1 - a_t)(F(w_t) - F^*)$$

$$+ \frac{a_t q_t c_t}{2\eta}\left(\|x^* - x_t\|^2 - \|x^* - x_{t+1}\|^2 - \|x_{t+1} - x_t\|^2\right) + \frac{La_t^2}{2}\|x_{t+1} - x_t\|^2$$

Dividing both sides by $a_t q_t c_t$ and summing up from $1$ to $T$, we have

$$\sum_{t=1}^{T} \frac{1}{a_t q_t c_t}(F(w_{t+1}) - F^*) \leq \sum_{t=1}^{T} \frac{1 - a_t}{a_t q_t c_t}(F(w_t) - F^*)$$

$$+ \sum_{t=1}^{T}\left(\frac{La_t}{2q_t c_t} - \frac{1}{2\eta}\right)\|x_{t+1} - x_t\|^2 + \frac{1}{2\eta}\|x^* - x_1\|^2$$

Note that $a_t \leq q_t$, $\frac{1}{a_t q_t c_t} \geq \frac{1 - a_{t+1}}{a_{t+1} q_{t+1} c_{t+1}}$ and $a_t = 1$. Thus

$$\frac{F(w_{T+1}) - F^*}{a_T q_T c_T} \leq \frac{\|x_1 - x^*\|^2}{2\eta} + \sum_{t=1}^{T}\left(\frac{L}{2b_t} - \frac{1}{2\eta}\right)\|x_{t+1} - x_t\|^2$$

$$= \frac{\|x_1 - x^*\|^2}{2\eta} + \sum_{t=1}^{T}\left(\frac{L}{2b_t} - \frac{1}{2\eta}\right)\frac{\eta^2 \|\nabla F(v_t)\|^2}{c_t^2 q_t^2}.$$

$\square$

## C.2 FIRST VARIANT

By using Lemma C.1, the proof idea of Theorem 5.1 is the same as the proof of Theorem 4.1. Hence, we omit it for brevity.

## C.3 SECOND VARIANT

By using Lemma C.1, the proof idea of Theorem 5.3 is the same as the proof of Theorem 4.3. Hence, we omit it here.

## C.4 A DISCUSSION ON WHEN $\Delta = 0$ AND $\delta = 1$

Algorithms 5 and 6 become one when $\Delta = 0$ and $\delta = 1$. As discussed in B.4, the challenge is to find a explicit bound on $b_T$. First, we give an asymptotic rate in Theorem C.2 of which the proof idea is the same as the proof of Theorem B.6, thus is omitted.

**Theorem C.2.** *Suppose $F$ satisfies Assumptions 1 and 2', when $\Delta = 0$ for Algorithm 5, or equivalently, $\delta = 1$ for Algorithm 6, by taking $a_t = \frac{2}{t+1}$, $p_t = \frac{2}{t}$, we have*

$$F(w_{T+1}) - F^* = O\left(1/T^2\right).$$

Now, we aim to prove the following non-asymptotic rate.

**Theorem C.3.** *Suppose $F$ satisfies Assumptions 1 and 2', when $\Delta = 0$ for Algorithm 5, or equivalently, $\delta = 1$ for Algorithm 6, by taking $a_t = \frac{2}{t+1}$, $p_t = \frac{2}{t}$, we have*

$$F(w_{T+1}) - F^* \leq \frac{4\left(b_0 + \frac{4\eta^2 L^2}{b_0}\right)\left(\frac{\|x_1 - x^*\|^2}{2\eta} + \frac{\eta^2 L}{b_0}\log^+ \frac{\eta L}{b_0}\right)}{T(T+1)} + \frac{16L\left(\frac{\|x_1 - x^*\|^2}{2\eta} + \frac{\eta^2 L}{b_0}\log^+ \frac{\eta L}{b_0}\right)^2}{T+1}.$$

We shortly discuss here why we can only give a rate in the order of $1/T$ but not $1/T^2$. Recall that in the proof of Theorem B.9, the key step is that after a certain time, $\|\nabla F(x_t)\|$ is a non-increasing sequence, by using which we can finally give a constant upper bound on $b_T$ that finally helps us to get the final $1/T$ rate. However, it is unclear under what condition on $b_t$, $\|\nabla F(v_t)\|$ now will be a non-increasing sequence in our accelerated algorithm. Thus it is unclear to us whether it is possible to give a constant bound on $b_T$. Instead, we will show $b_t$ can increase at most linearly in this accelerated scheme by a new trick, for which reason, we can finally obtain the rate in the order of $1/T$. This guarantees that the convergence of the last iterate is no worse than the variants in Section 4.

*Proof.* As before, to start with, we use Lemma C.1

$$\frac{F(w_{T+1}) - F^*}{a_T q_T b_T} \leq \frac{\|x_1 - x^*\|^2}{2\eta} + \sum_{t=1}^{T}\left(\frac{L}{2b_t} - \frac{1}{2\eta}\right)\frac{\eta^2 \|\nabla F(v_t)\|^2}{b_t^2 q_t^2}.$$

By using $b_t^2 = b_{t-1}^2 + \frac{\|\nabla F(v_t)\|^2}{b_t^2}$ and the same technique in the previous proof, we know

$$\sum_{t=1}^{T}\left(\frac{L}{2b_t} - \frac{1}{2\eta}\right)\frac{\eta^2 \|\nabla F(v_t)\|^2}{b_t^2 q_t^2} \leq \frac{\eta^2 L}{b_0}\log^+ \frac{\eta L}{b_0}.$$

So we have

$$F(w_{T+1}) - F^* \leq a_T q_T b_T \underbrace{\left(\frac{\|x_1 - x^*\|^2}{2\eta} + \frac{\eta^2 L}{b_0}\log^+ \frac{\eta L}{b_0}\right)}_{D}$$

Now we turn to bound $b_t$ by observing

$$\begin{aligned}
b_t^2 &= b_{t-1}^2 + \frac{\|\nabla F(v_t)\|^2}{q_t^2} \\
&\leq b_{t-1}^2 + \frac{2\|\nabla F(v_t) - \nabla F(w_{t+1})\|^2}{q_t^2} + \frac{2\|\nabla F(w_{t+1})\|^2}{q_t^2} \\
&\leq b_{t-1}^2 + \frac{2L^2\|v_t - w_{t+1}\|^2}{q_t^2} + \frac{2\|\nabla F(w_{t+1})\|^2}{q_t^2} \\
&= b_{t-1}^2 + \frac{2L^2 a_t^2\|x_{t+1} - x_t\|^2}{q_t^2} + \frac{2\|\nabla F(w_{t+1})\|^2}{q_t^2} \\
&\leq b_{t-1}^2 + 2L^2\|x_{t+1} - x_t\|^2 + \frac{2\|\nabla F(w_{t+1})\|^2}{q_t^2},
\end{aligned}$$

where the last inequality is due to $a_t \leq p_t$. Then we use $\|x_{t+1} - x_t\|^2 = \frac{\eta^2 \|\nabla F(v_t)\|^2}{b_t^2 q_t^2} = \frac{\eta^2 (b_t^2 - b_{t-1}^2)}{b_t^2}$ and $\|\nabla F(w_{t+1})\|^2 \leq 2L(F(w_{t+1}) - F^*) \leq 2La_t q_t b_t D$ to get

$$b_t^2 \leq b_{t-1}^2 + \frac{2\eta^2 L^2 \left(b_t^2 - b_{t-1}^2\right)}{b_t^2} + \frac{4La_t q_t b_t D}{q_t^2}$$

$$\leq b_{t-1}^2 + \frac{2\eta^2 L^2 \left(b_t^2 - b_{t-1}^2\right)}{b_t^2} + 4Lb_t D$$

$$\Rightarrow b_t \leq \frac{b_{t-1}^2}{b_t} + 2\eta^2 L^2 \frac{b_t^2 - b_{t-1}^2}{b_t^3} + 4LD$$

$$\leq b_{t-1} + 4\eta^2 L^2 \left(\frac{1}{b_{t-1}} - \frac{1}{b_t}\right) + 4LD$$

$$\Rightarrow b_t \leq b_0 + \frac{4\eta^2 L^2}{b_0} + 4LDt.$$

Using this bound, we finally get

$$F(w_{T+1}) - F^* \leq a_T q_T b_T D$$

$$= \frac{4 \left(b_0 + \frac{4\eta^2 L^2}{b_0} + 4LDT\right) D}{T(T+1)}$$

$$= \frac{4 \left(b_0 + \frac{4\eta^2 L^2}{b_0}\right) \left(\frac{\|x_1 - x^*\|^2}{2\eta} + \frac{\eta^2 L}{b_0} \log^+ \frac{\eta L}{b_0}\right)}{T(T+1)} + \frac{16L \left(\frac{\|x_1 - x^*\|^2}{2\eta} + \frac{\eta^2 L}{b_0} \log^+ \frac{\eta L}{b_0}\right)^2}{T+1}.$$

$\square$

## D EXPERIMENTS

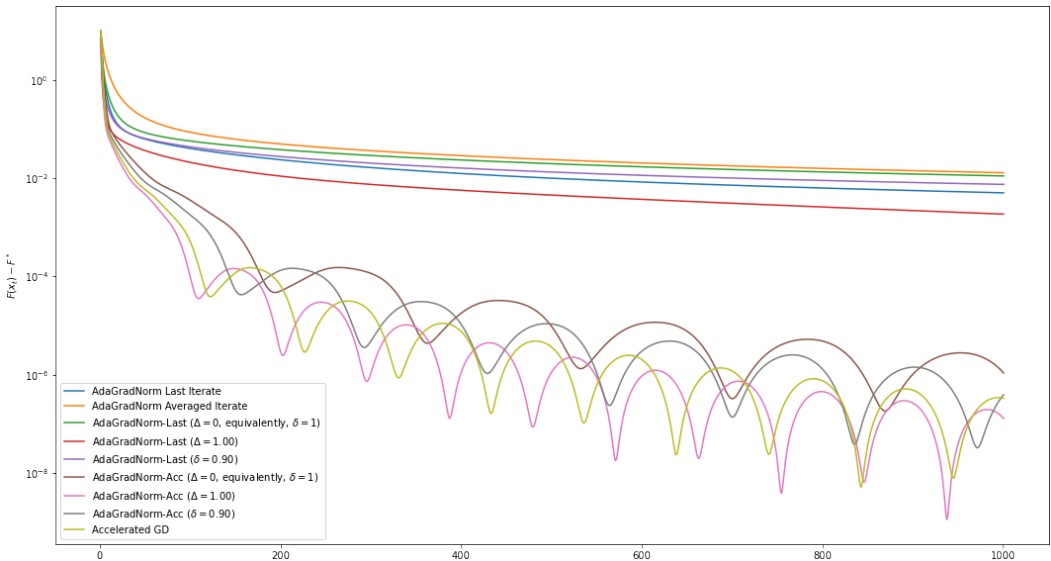

Figure 1: Function value gap for different algorithms

In this section, we provide some empirical evidence to compare the performances of our algorithms in the deterministic setting. Our test function follows the quadratic function used to prove the lower bound of the first order method constructed by Nesterov (Nesterov et al., 2018). That is

$$F(x) = \frac{x[1]^2 + x[d]^2 + \sum_{i=1}^{d-1} \left(x[i] - x[i+1]\right)^2}{2} - x[1].$$

where $x[i]$ refers to the $i$-th coordinate of point $x \in \mathbb{R}^d$. It is known that $F$ is 4-smooth and convex with the unique minimizer

$$x^*[i] = 1 - \frac{i}{d+1}, \forall i \in [d].$$

We fix $d = 101$ and set the time horizon to $T = 1000$ in the test. The starting point $x_1$ is initialized randomly satisfying that every coordinate is uniformly chosen in $[0, 1)$. All algorithms share the same $x_1$. For the adaptive algorithms, we choose $b_0 = 10^{-2}$ and set $\eta = 1$ without any further tuning. We also compare with an accelerated algorithm (Lan, 2020), which requires using the smoothness constant $L = 4$.

The result is shown in Figure 1. We can find that our Algorithms 3 and 4 admit the last iterate convergence. Additionally, both our accelerated algorithms, i.e., Algorithms 5 and 6, enjoy the accelerated property without knowing the smoothness parameter and are competitive against Accelerated Gradient Descent (Lan, 2020) which requires the smooth parameter to set the step size. Another interesting observation is that it seems AdaGradNorm also exhibits the last iterate convergence. However, whether this is indeed a property of AdaGradNorm has not been confirmed by the theory. We leave this as a future direction.

