# OpenReview forum: "On the Convergence of AdaGrad(Norm) on $\mathbb{R}^d$: Beyond Convexity, Non-Asymptotic Rate and Acceleration"
_ICLR.cc/2023/Conference — ICLR 2023 poster_

### Official Review · Reviewer_qVCv · 2022-10-23

**Confidence:** 4
**Correctness:** 4
**Technical Novelty And Significance:** 2
**Empirical Novelty And Significance:** Not applicable
**Recommendation:** 6

**Clarity, Quality, Novelty And Reproducibility:**

**Clarity**

The writing of this paper is clear and easy to understand.

**Novelty**

To the best of my knowledge, there are no such results in the existing literature. However, the novelty can be further improved as discussed in the "Strength And Weaknesses" part.

**Strength And Weaknesses:**

**Strength**

1. The presentation of this paper is clear and it is easy to understand the results of this paper and the proof technique.

**Weakness**

1. I am not sure whether understanding the convergence of AdaGrad in the convex regime is an important question. First, to the best of my knowledge, AdaGrad is seldomly adopted in machine learning practice, especially in deep learning. Second, I think that it is more important to study the non-convex setting.

2. The current logic and organization of this paper need to be improved. The current organization of this paper seems to "collect" all the results. For example, AdaGrad-Norm Last is proposed to ensure the last iteration convergence of AdaGrad-Norm. But does AdaGrad-Norm really fail to converge in terms of the last iteration or it is just due to the limitation in proof? Discussion is needed here.

3. While removing the bounded domain assumption, the result for the stochastic case is still somewhat restricted. For convex objectives, SGD is known to converge with the bounded noise variance assumption, which is strictly weaker than the sub-gaussian noise assumption. Can similar results be derived for AdaGrad-Norm?

4. While Section 1.1 claims "show an explicit non-asymptotic convergence rate of AdaGrad-Norm and AdaGrad on $R^d$ in the deterministic setting", I only find such a result for AdaGrad-Norm (Theorem 3.1). Did I miss something?

5. I feel that the authors need to include some experiments to justify the usefulness of the proposed AdaGrad-Norm variants. For example, does AdaGrad-Norm-Acc really converge faster than AdaGrad-Norm over convex objectives? Also, what is the advantage of AdaGrad-Norm-Acc over Nesterov's momentum given that they can both achieve the lower bound of the convergence rate?

6. There are missing related works. Faw et al. (2022) derive the convergence of AdaGrad-Norm in the non-convex setting with the least assumptions. They also provide detailed references to the works studying the convergence of AdaGrad.

**References**

Faw et al., The Power of Adaptivity in SGD: Self-Tuning Step Sizes with Unbounded Gradients and Affine Variance, 2022

**Summary Of The Paper:**

This paper studies the convergence of AdaGrad and its variants over the convex objectives. Specifically, the authors first prove the convergence of AdaGrad-Norm both in the deterministic case and in the stochastic case. Compared to existing results, the derived results remove the bounded domain assumption. The authors then propose two types of variants of AdaGrad. The first type can ensure the last iterate convergence in the deterministic case and the second can achieve a faster convergence rate.

**Summary Of The Review:**

Despite that the results do improve those in the existing literature, I find that the studied problem is less interesting, the theoretical results may be further improved, and the proposed algorithms need to be justified in more detail.

---

### Official Review · Reviewer_sbsF · 2022-10-25

**Confidence:** 4
**Clarity, Quality, Novelty And Reproducibility:** The paper is a little unclear, very n…
**Correctness:** 3
**Technical Novelty And Significance:** 3
**Empirical Novelty And Significance:** Not applicable
**Recommendation:** 6

**Strength And Weaknesses:**

Strength:
  1. The proposed analysis techniques are, as far as I am aware, novel and interesting
  2. The authors have formulated the analysis in a way that is easy to understand their contributions

Weaknesses:

  Before I start talking about the weakness, I want to emphasize that, as a theory person, I highly appreciate the authors for trying to figure out different ways to provide the convergence analysis of algorithms. Although I don't really have enough time to read all the proof details, the analysis that the authors try to provide are very interesting and potentially impactful.

  However, I have the feeling that the paper is a bit over-selling, or the results presented are not really matched with the authors' claims. Most of the paper is concentrated on a variant of AdaGrad, AdaGradNorm, which I admit that I am not familiar with. I think it is not very popular, and the analysis techniques that the authors have used for this algorithm, maynot be easily transferrable to the vanilla AdaGrad algorithm. I tried to read some parts of the Appendix as well, and it seems that the only setting that the authors have provided analysis on AdaGrad is, the deterministic setting (Sec A.3). The stochastic version of AdaGrad is not analyzed.

  Moreover, the nice properties and the variants are all proposed based on the AdaGradNorm algorithm, instead of the original AdaGrad algorithm. Can we propose similar variants for AdaGrad and get similar properties? This makes it also hard to interpret the results as a more advanced/detailed analysis of AdaGrad, but rather AdaGradNorm. Therefore, I feel the results in this paper are important, but the way the authors sell them are not completely aligned with the results.

  Finally, the results are established under the quasar-convex setting, and I wonder whether they can be further extended to the nonconvex setting. (I have a very strong feeling that this can be done easily since in the nonconvex setting, similar things such as the telescopic sums in Sec 3.1 appear in the analysis too). Some (simple and synthetic) experimental results that support the correctness of the theorems will be highly appreciated.


*After Rebuttal*

I thank the authors for a nice rebuttal. I have increased my rating.

**Summary Of The Paper:**

This paper introduces some new analysis technique of the convergence of AdaGrad on the unbounded domain, in the smooth convex optimization setting. The authors prove that several variants of AdaGrad can converge (in both average and the last literature sense) to the optimum without the need of additional assumptions.

**Summary Of The Review:**

In summary, I do like the idea of the paper as a theory person. However, I just feel like it should be further improved in its current form.

I highly appreciate the authors for trying to find new ways to prove the convergence of AdaGrad, and its variants, and I believe the results could be very significant. Extending the current results to, e.g., stochastic AdaGrad, last iterate convergence of AdaGrad would be much more interesting than dealing with the variant AdaGradNorm. Or extending the results to the nonconvex setting would be very interesting.

That being said, the above suggestions may make the paper a bit too-long and too technical for a (mostly deep-learning) conference like ICLR, I would suggest re-submitting to ML journals like JMLR, or theory conferences like COLT.

I am happy to change my rating if the authors can convince me of their contributions.

---

### Official Review · Reviewer_iRiy · 2022-10-26

**Confidence:** 4
**Correctness:** 4
**Technical Novelty And Significance:** 3
**Empirical Novelty And Significance:** Not applicable
**Recommendation:** 8

**Clarity, Quality, Novelty And Reproducibility:**

Although your title is about analyzing AdaGrad the analysis is actually for AdaGradNorm. I suggest either changing the title or extending your analysis for the vanilla/general AdaGrad.

**Strength And Weaknesses:**

As much as it is known this is the first analysis that extends the convergence analysis of Adagrad to the unconstrained domain.

**Summary Of The Paper:**

This paper provides a theoretical analysis of AdaGradNorm which is a variant of AdaGrad in the unconstrained domain. As much as it is known this is the first analysis that extends the convergence analysis of Adagrad to the unconstrained domain. They analyze both deterministic and stochastic settings. The analysis is provided for the quasar-convex that is bigger than the class of convex functions and includes it. Finally, they also propose accelerated variants of AdaGradNorm in the deterministic setting with convergence analysis which indicates the acceleration occurs compared to Non-accelerated variants.


**Summary Of The Review:**


Comments:
1- As an empirical justification, it would be useful to include some empirical results for your accelerated variant in the paper.

2- Although your title is about analyzing AdaGrad the analysis is actually for AdaGradNorm. I suggest either changing the title or extending your analysis for the vanilla/general AdaGrad.

Minor comments:
1- Adding numbers for your important equations would be helpful.
2- In the 2nd last eq in page 6, I guess a \sum is missing.

---

### Public Comment · ~Kyunghun_Nam1 · 2023-09-30
**Inquiry about your paper**

Dear
I hope this message finds you well. I would like to address specific queries.

In Appendix A.2.1 (page 14), the proof of Lemma 3.4, I can't understand how the following equality is hold.

$\mathbb{E} \left[ \frac{\langle - \xi_t, x_t - x^{\star} \rangle}{\gamma (2 b_t - b_0)} \right] = \mathbb{E} \left[ \frac{1}{\gamma} \left( \frac{1}{2b_t - b_0} - \frac{1}{2b_{t-1} - b_0} \right) \langle - \xi_t, x_t - x^{\star} \rangle \right]$


Can you kindly explain my question, please?

Thanks
Warm regards

---

> ### Author Response · Authors · 2023-10-01
> **Response**
>
> Thanks for your interest in our paper.
>
> The equation holds due to $\xi_t$ being independent of the history up to $x_t$. Formally speaking, let $H_t=\\{x_1,\xi_1,\cdots,x_{t-1}, \xi_{t-1}, x_t\\}$, there is
>
> $\begin{align*} \mathbb{E}[\langle \xi_t, x_t-x^* \rangle/(2b_{t-1}-b_0)]=\mathbb{E}[\mathbb{E}[\langle \xi_t, x_t-x^* \rangle/(2b_{t-1}-b_0)\mid H_t]]=\mathbb{E}[\langle \mathbb{E}[\xi_t\mid H_t], x_t-x^* \rangle/(2b_{t-1}-b_0)]=0,\end{align*}$ where the first equation is by tower property. Therefore, the equation in your question is true.
>
> Best,
>
> The Authors.

---

### Author Response · Authors · 2023-10-05
**Updated Manuscript**

To Whom It May Concern,

Thanks for being interested in our paper. We provide an updated version on the arxiv with fixed typos: https://arxiv.org/abs/2209.14827.

Best regards,

The Authors

---

### Decision · Program_Chairs · 2023-01-20

**Decision:**

Accept: poster

**Justification For Why Not Higher Score:**

Even though adaptive methods are of general interest to the ML community, the paper's main contribution is somewhat narrow in scope so it does not justify a higher degree of exposure.

**Justification For Why Not Lower Score:**

The paper presents a solid contribution in the literature on adaptive methods.

**Metareview: Summary, Strengths And Weaknesses:**

This paper studies the rate of convergence of AdaGrad (with scalar step-size adaptation) in both stochastic and deterministic (unconstrained) settings. Compared to previous work by Li & Orabona and Antonakopoulos et al., the authors provide an non-asymptotic convergence rate guarantee which gives explicit values to some of the hidden - and possibly large - constants presented in previous works. The authors likewise provide an accelerated variant of AdaGrad which closes a gap in the study of adaptive methods - like AcceleGrad, UnderGrad, and the like - which only apply to problems with a bounded feasible region (or knowledge of a bounded set containing a solution of the problem).

The reviewers were overall positive regarding the paper, and even though it was suggested that a more theoretical venue like COLT or ALT might be more appropriate, the interest of the ML community at large in adaptive methods justifies the paper's inclusion in ICLR, so there were no objections to an "accept" decision.

**Note From Pc:**

if the above contains the word "oral" or "spotlight" please see: "oral" presentation means -> notable-top-5% and "spotlight" means -> notable-top-25%. As stated in our emails, we are disassociating presentation type from AC recommendations

**Summary Of Ac-Reviewer Meeting:**

This was not a borderline paper.